# 🎧 SmartDJ: Declarative Audio Editing with Audio Language Model

**Zitong Lan, Yiduo Hao, Mingmin Zhao**
University of Pennsylvania
`https://waves.seas.upenn.edu/projects/smartdj`

## Abstract

Audio editing plays a crucial role in VR/AR immersion, virtual conferencing, sound design, and interactive media. However, recent generative audio editing models depend on template-like instruction formats and are restricted to mono-channel audio. Moreover, existing systems require users to specify low-level editing actions, rather than expressing the desired outcome at a higher semantic level. We introduce SmartDJ, a novel framework for stereo audio editing that enables declarative audio editing, where the users describe the desired outcome while delegating the underlying editing operations to the system. Given a high-level instruction, SmartDJ decomposes it into a sequence of atomic edit operations, such as adding, removing, or spatially relocating sound events. These operations are then executed by a diffusion model trained to edit stereo audio. To enable this capability, we design a scalable data synthesis pipeline that produces paired examples of declarative instructions, atomic edit operations, and audios before and after each edit operation. Experiments demonstrate that SmartDJ achieves superior perceptual quality, spatial realism, and semantic alignment compared to prior audio editing methods.

## 1 Introduction

Imagine you recorded a sound in forest on a rainy day and wanted to transform it into the soundscape of a sunny forest. Achieving this transformation requires many edits: removing elements like rainfall and adding new effects such as leaf rustling. Traditionally, audio editing is a *procedural process*: the user specifies how to achieve the goal, step by step, by removing rain, layering new samples, adjusting gain, and so on. Yet in practice, users would prefer just to issue a single, declarative instruction, e.g., *"make this sound like a sunny day"*, and rely on an intelligent audio editor to decide what edits are needed. We call this *declarative editing*: the user declares *what* the desired outcome should be, while leaving the *how*, i.e., the sequence of operations, to the system. Such a declarative paradigm would unlock a wide range of applications in VR/AR, gaming, cinematic post-production, and beyond, where designing and modifying audio scenes is central to immersive experiences.

Recent advances in deep generative models have opened exciting possibilities for text-to-audio generation and language-driven editing (Evans et al., 2024a;b; Hai et al., 2024; Heydari et al., 2024; Huang et al., 2023a; Liu et al., 2023a; 2024a; Jia et al., 2024; Liang et al., 2024; Wang et al., 2023; Xu et al., 2024). However, existing audio editors remain limited in two key aspects. First, they rely on templated instructions such as "add the sound of birds" or "remove the sound of rain", which restricts their ability to handle declarative or abstract user instructions. Second, they operate only on monaural (single-channel) audio, discarding spatial features such as interaural time and level differences, which are primary cues for spatial hearing. Without these cues, even semantically correct edits sound flat and fail to deliver immersive experiences.

In contrast, declarative editing requires an editor to bridge from declarative goals to detailed operations automatically. When a user issues instructions such as *"place me in a concert hall"* or *"have this audio in a library"*, the system must reason about which sounds to remove, which to preserve, how to adjust loudness, when to introduce new events, and how to shift spatial position. Current audio editors based on diffusion models cannot achieve this, as they lack the reasoning capacity to interpret these declarative instructions. On the other hand, language models alone are also insufficient: while they can parse the user's request, they lack grounding in the audio itself, making it impossible to

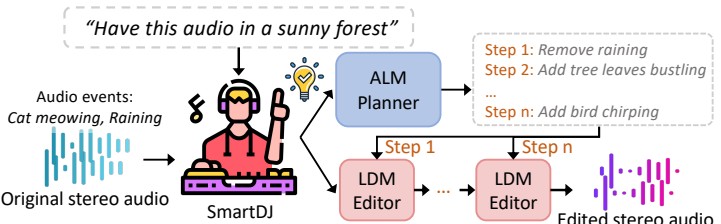

Figure 1: SmartDJ has an Audio Language Model (ALM) that acts as an edit planner to decompose the instructions into atomic steps, guiding the Latent Diffusion Model (LDM) editor to produce high-quality edited audio.

decide which sound elements should be suppressed, emphasized, or retained. This gap highlights the need for a framework that jointly reasons over language and audio, combining natural-language instruction understanding with audio-aware analysis and editing.

In this work, we present SmartDJ, the first framework for declarative audio editing by introducing Audio Language Models (ALM) into the audio editing loop. This approach is motivated by the recent success of MLLMs in multimodal grounding and reasoning (Chu et al., 2023; Ghosh et al., 2025a; Kong et al., 2024; Liu et al., 2023b; Li et al., 2023a). As illustrate in Fig. 1, our key idea is to let the ALM act as a *planner*: it perceives the original audio while interpreting the user's declarative instruction and decomposes it into *a sequence of atomic edit operations*, such as adding or removing a sound, shifting direction, or adjusting volume, etc. These atomic edit operations are then executed sequentially by a conditional Latent Diffusion Model (LDM) as an audio *editor*, realizing the user's declarative goal. This separation of planning and editing transforms audio editing from a procedural task into a declarative one. Moreover, because the intermediate representation is expressed in natural language, users can easily inspect, refine, or override the planned edits interactively.

Training a system to perform declarative editing introduces a fundamental challenge: it requires paired examples of *declarative instructions*, their corresponding *atomic edit sequences*, and the *before-and-after edited audio*. Such data is rarely available, as natural soundscapes are complex and difficult to edit at scale. To address this, we construct a scalable data generation pipeline that provides *controllable audio scenes*. Each scene is assembled from independently parameterized sound events with attributes such as direction and loudness. Within this pipeline, an off-the-shelf LLM acts as the *designer*, producing diverse declarative editing instructions and the corresponding atomic operations. Audio signal processing then serves as the *composer*, rendering each operation by adjusting or mixing sound events to produce the corresponding audio after every edit. Together, this designer–composer pipeline mirrors the planner–editor structure of our model. It produces a large-scale corpus of realistic editing operations, providing the supervision needed to train and evaluate our model.

Experimental results show that SmartDJ delivers superior editing quality and better alignment with declarative user instructions from both subjective metrics and human evaluations. Our LDM also outperforms existing baselines for audio editing. Ablation studies show that our ALM can effectively reason about and decompose declarative user instructions into a sequence of editing actions.

In summary, our main contributions are as follows:

- We introduce SmartDJ, the first stereo audio editor capable of interpreting declarative user instructions with an ALM and executing them as precise atomic edit operations through an LDM.
- We introduce the first scalable pipeline for generating editable stereo audio scenes, combining declarative instructions with controllable events to enable reasoning-based audio editors.
- We conduct extensive experiments and user studies with different baseline methods and demonstrate that SmartDJ achieves the highest editing quality for both objective and subjective metrics.

## 2 RELATED WORK

**Audio generation and editing.** With the current advances in deep generative models, lots of methods have achieved high-quality audio generation from text and multi-modal conditions (Chen et al., 2024; Evans et al., 2024a; Hai et al., 2024; Huang et al., 2023a; Liu et al., 2023a; 2024a; Wang et al., 2023). Recently, spatial audio generation has attracted more attention (Evans et al., 2024a; Heydari et al., 2024; Liu et al., 2025; Sun et al., 2024), providing more realistic and immersive listening experiences. Parallel to these generation efforts, text-guided audio editing also emerged as a powerful tool for

modifying existing audio recordings. Audit (Wang et al., 2023) introduced an end-to-end diffusion model conditioned on both the input audio and simple, structured text commands, but its reliance on fixed editing templates limits flexibility to interpret declarative user prompts. WavCraft (Liang et al., 2024) leverages GPT-api to parse user instructions, yet it expects fully specified prompts (e.g., "extract baby crying from the audio" or "apply a low-pass filter to the wave crashing sound and add it back"). Some recent work (Jia et al., 2024; Manor & Michaeli, 2024; Xu et al., 2024) adapts image-editing techniques (e.g., DDPM inversion, Null-text inversions, attention map manipulation) to monaural audio. They require precise token-level modification to complete the edit, struggle with declarative instructions. To summarize, none of the existing work can interpret declarative user input to complete the audio editing. Besides, existing frameworks remain confined to monaural outputs and are ill-suited for immersive spatial scenarios Lan et al. (2024; 2025a;b).

**Multimodal Large Language Model.** Large language models (LLMs) are remarkable in natural language processing tasks. Provided with multimodal inputs, such as images and audio, multimodal LLMs (MLLMs) (et al., 2022; Liu et al., 2023b; Li et al., 2023a; Team, 2025; Kong et al., 2024b; Ghosh et al., 2025b) demonstrate exceptional performance across a wide range of downstream visual-language and audio-language tasks. In the vision domain, LLaVA (Liu et al., 2023b) enables LLMs to achieve general-purpose visual and language understanding by fine-tuning on a multimodal instruction-following dataset. In the audio domain, LTU (Gong et al., 2023) and Audio Flamingo (Kong et al., 2024b; Ghosh et al., 2025b) enhance LLMs with the ability to process non-speech sounds and non-verbal speech. With strong capabilities of MLLMs, researchers introduced them into the field of visual content generation (Wu et al., 2024c; Xia et al., 2023; Wu et al., 2024b; Koh et al., 2023; Fu et al., 2024; Huang et al., 2023b), world modeling (Wu et al., 2024a; Ge et al., 2024; Mai et al., 2024), grounding (Lai et al., 2024; Hao et al., 2024; Cheng et al., 2024), and embodied AI (et. al, 2023; Li et al., 2023b; Mu et al., 2023; Szot et al., 2024; Wu et al., 2025). In image generation and editing, various works (Koh et al., 2023; Fu et al., 2024; Huang et al., 2023b) use VLM to guide diffusion models. However, in the audio domain, existing methods have not exploited the reasoning capabilities of audio language models.

**Diffusion-based Image Editing.** In contrast to text-to-image generation, image editing focuses on altering specific elements or attributes within an image while preserving the contents of the remaining image. Diffusion models have been widely used in image editing tasks (Couairon et al., 2022; Hertz et al., 2022; Hui et al., 2024) by altering the inversion process, which produces a latent representation that can reconstruct the image through the generative process. SDEdit (Meng et al., 2022) first adds noise to the source image, and then subsequently denoises the image through the SDE to produce the target image. P2P (Hertz et al., 2022) adjusts the cross-attention features according to the difference between the source and target captions to generate the target images. Based on this, IP2P (Brooks et al., 2023) finetuned a diffusion model on edit image triplets to enable image editing with simple natural language instructions. Following works (Geng et al., 2023; Hui et al., 2024) further scale up the dataset to support more capable and generalized models. Furthermore, later works (Fu et al., 2024; Huang et al., 2023b) use vision-language models to guide diffusion models for image editing.

## 3 METHOD

In this section, we first define the task and notations. We then introduce our proposed framework SmartDJ that combines an Audio Language Model for interpreting declarative editing instructions with a diffusion model for executing sequential audio edits. Finally, we describe a scalable data generation pipeline powered by LLMs to support supervised training and evaluation.

### 3.1 PROBLEM DEFINITION AND NOTATIONS

Let $a_0$ denote the original audio waveform, which contains multiple audio events (e.g., *cat meowing*, *rainfall*), as shown in Fig. 1. We define a *declarative editing instruction* $\mathcal{P}$ as a natural language description of a desired transformation of the overall audio scene. Such instructions are typically declarative: they specify what the desired outcome should be (e.g., "*make it sound like a quiet morning in a sunny forest*" or "*transform this into an indoor library setting*"), but do not explicitly prescribe the individual operations. Since $\mathcal{P}$ is a declarative instruction, it must be decomposed into a detailed sequence of atomic editing steps $\mathcal{S} = \{s_1, s_2, \ldots, s_n\}$, where each step $s_i$ either modifies an existing audio event in $a_0$ or introduces a new event needed to fulfill $\mathcal{P}$. We denote $a_i (i = 1, 2, ..., n)$ as the intermediate audio after applying step $s_i$, where $a_0$ is the original input and $a_n$ is the final edited audio.

Specifically, each step $s_i$ either modifies an existing audio event or introduces a new event required to satisfy $\mathcal{P}$. The atomic editing operations considered in this work are:

- `Add:` Mix a new sound event into the scene (e.g., inject bird chirps).
- `Remove:` Delete an existing sound event (e.g., remove car engine noise).
- `Extract:` Isolate a particular sound event from the original audio while removing others.
- `Volume adjust:` Adjust the volume of a specific event.
- `Change direction:` Modify the spatial location of an event.
- `Time shift:` Adjust the sound timing by x seconds.
- `Add reverberation:` Add reverberation to a single source.
- `Timbre adjust:` Modify the timbre of a single sound source.

The goal of the editing is to produce a target edited audio clip $a_n$ by applying the sequence of edits $\mathcal{S}$ to $a_0$. Importantly, this editing formulation should preclude shortcut solutions that ignore the original input (e.g., re-generating an entirely new audio from scratch). Instead, the edited audio $a_n$ must preserve all unedited content from $a_0$ while achieving the requested audio scene transformation. Please refer to Appendix A.3 for more details.

## 3.2 SMARTDJ FRAMEWORK

SmartDJ consists of an Audio Language Model (ALM) and a Latent Diffusion Model (LDM). We leverage the ALM as a planner to interpret declarative instructions and generate a sequence of atomic editing steps. The LDM editor then executes these steps sequentially to transform the original audio. As illustrated in Fig. 2, the ALM takes the original audio $a_0$ and the declarative editing instruction $\mathcal{P}$ as input, and outputs a sequence of atomic editing steps $\mathcal{S} = \{s_1, s_2, ..., s_n\}$. These editing steps are then executed sequentially by the LDM, producing intermediate results $a_1, a_2, \ldots, a_n$, where $a_n$ is the final edited audio. The overall process is formulated as:

$$\{s_1, s_2, ..., s_n\} = \text{ALM}(a_0; \mathcal{P}) \tag{1}$$

$$a_i = \text{LDM}(a_{i-1}; s_i), \, i = 1, 2, ..., n \tag{2}$$

Figure 2: SmartDJ framework overview

## 3.3 AUDIO LANGUAGE MODEL FOR ATOMIC EDITING STEPS GENERATION

The Audio Language Model (ALM) takes as input the original audio clip and the declarative editing instruction and generates a sequence of atomic editing steps. As shown at the top of Fig. 3, we first encode $a_0$ using a pretrained audio encoder (i.e., CLAP (Wu et al., 2023)) to obtain an audio embedding $z_a$, which is injected into the ALM via adapter layers. In parallel, the instruction $\mathcal{P}$ is tokenized and encoded as a sequence of embeddings $(p_1, p_2, \ldots, p_k)$, which serve as the textual context for the ALM.

Our ALM is trained in an auto-regressive manner to generate the token sequence corresponding to the atomic editing steps $\mathcal{S}$, by minimizing the following objective:

$$\mathcal{L}_{\text{ALM}} = -\sum_{t=1}^{l} \log P_\theta(r'_t = r_t \mid z_a, r_{1:t-1}, p_{1:k}), \tag{3}$$

where $r$ and $r'$ are the ground truth and predicted text tokens for the atomic editing steps $\mathcal{S}$, $l$ is the length of the tokens, and $\theta$ denotes the model parameters. To enable efficient fine-tuning, we freeze the parameters of the CLAP audio encoder, apply Low-Rank Adaptation (LoRA) (Hu et al., 2022) to a small subset of the LLM layers(Ghosh et al., 2025b), and fully fine-tune the adapter layers.

**Separate training.** We train the ALM and LDM as independent modules rather than end-to-end. This enables human-in-the-loop editing, where users can easily intervene at the level of generated natural-language-based atomic steps before LDM editor inference. Besides, it makes training and computation more efficient and promotes modularity. This design allows for different ALMs or LDMs to be swapped in or out with minimal re-training effort, making it both practical and extensible for diverse editing scenarios.

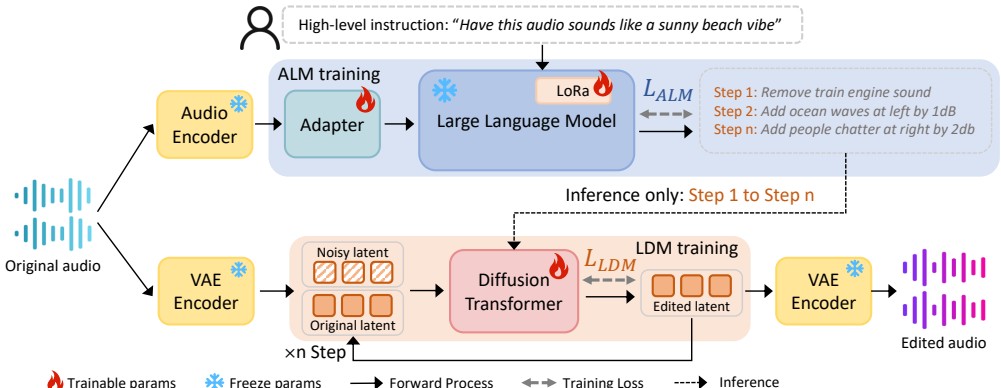

Figure 3: **SmartDJ framework**. Our method incorporates an ALM as an edit planner that understands both the original audio and the declarative instructions to produce atomic edit steps. These atomic steps are then fed into an LDM editor to edit the audio sequentially. The ALM and LDM modules are trained separately.

### 3.4 SEQUENTIAL STEREO AUDIO EDITING WITH LATENT DIFFUSION MODEL

The Latent Diffusion Model (LDM) in our framework performs audio editing conditioned on the atomic editing steps $\mathcal{S}$. To support this, we adopt a latent diffusion architecture (Hai et al., 2024; Rombach et al., 2022) and extend it to enable editing of stereo audio with spatial effects.

**Stereo Audio VAE.** Given a stereo audio signal $a \in \mathbb{R}^{2 \times L}$, where $L$ is the number of time-domain samples at two audio channels (left and right). The audio Variational Autoencoder (VAE) encodes $a$ into a latent representation $\hat{a} \in \mathbb{R}^{C \times L'}$, where $C$ and $L'$ denote the number of latent channels and the temporal length of the latent sequence. Similar to DAC (Kumar et al., 2023) and Stable-Audio-Open (Evans et al., 2024b), our audio VAE is based on a 1D-CNN autoencoder with a continuous VAE bottleneck and snake activation functions. The resulting latent $\hat{a}$ has a dimension of $C = 128$ and length of $L' = L/480$, resulting a compression ratio of $7.5 \times$.

**Latent Diffusion Model:** Our diffusion model conditions on both the text description $s_i$ at the $i$-th editing step and the latent representation of the audio from the previous step, $\hat{a}_{i-1}$, to generate the updated latent $\hat{a}_i$. We use the FLAN-T5 (Chung et al., 2024) text encoder $E_{\text{text}}()$ to convert $s_i$ to text embeddings. At each editing step, we initialize a randomly noised latent $\hat{a}_i' \in \mathbb{R}^{C \times L'}$, which is concatenated with $\hat{a}_{i-1}$ to form the input $[\hat{a}_{i-1}; \hat{a}_i'] \in \mathbb{R}^{2C \times L'}$ to the diffusion model. The model is conditioned on $E_{\text{text}}(s_i)$ via cross-attention layers, and the diffusion timestep $t$ is incorporated through a modified AdaLN module (Hai et al., 2024) to reduce model parameters. We implement a Diffusion Transformer (DiT) that learns to denoise the latent $\hat{a}_i'$ across multiple timesteps by predicting the added noise. Let $\epsilon$ denote the true added Gaussian noise, and let $\epsilon_\theta(\cdot)$ be the predicted noise output by the model. The training objective is to minimize the following denoising loss:

$$\mathcal{L}_{\text{LDM}} = \mathbb{E}_{\epsilon \sim \mathcal{N}(0,I), t, s_i, \hat{a}_{i-1}, \hat{a}_i'} \left\| \epsilon - \epsilon_\theta(t, E_{\text{text}}(s_i), [\hat{a}_{i-1}; \hat{a}_i']) \right\|_2. \quad (4)$$

During inference, we use DDIM sampling (Song et al., 2021) with classifier-free guidance (CFG), which has proven effective for text-guided generation and editing (Ho & Salimans, 2022). CFG steers the denoising process by interpolating between conditional and unconditional model predictions:

$$\tilde{\epsilon}_\theta = \omega \cdot \epsilon_\theta(t, E_{\text{text}}(s_i), [\hat{a}_{i-1}; \hat{a}_i']) + (1 - \omega) \cdot \epsilon_\theta(t, \varnothing, [\hat{a}_{i-1}; \hat{a}_i']), \quad (5)$$

where $\omega$ is the guidance scale and $\varnothing$ denotes the text embedding of an empty string.

### 3.5 DECLARATIVE INSTRUCTION AUDIO EDITING DATASET CURATION

Since no public dataset features audio editing conditioned on declarative instructions, we develop a scalable data generation pipeline, as illustrated in Fig. 4a. For each data point, we first randomly sample $K$ single-event audio clips from public datasets, each labeled with tags such as {"*car engine*", "*bell ring*", "*goat bleat*", ...}. We feed these labels into GPT-4o and prompt it to act as a sound designer: it designs a declarative editing instruction $\mathcal{P}$ that transforms the original mix into a new audio scene (e.g., *"Make this sound like a countryside morning"* or *"Make it sound like a busy train station on a sunny afternoon"*). It then decomposes $\mathcal{P}$ into a sequence of atomic

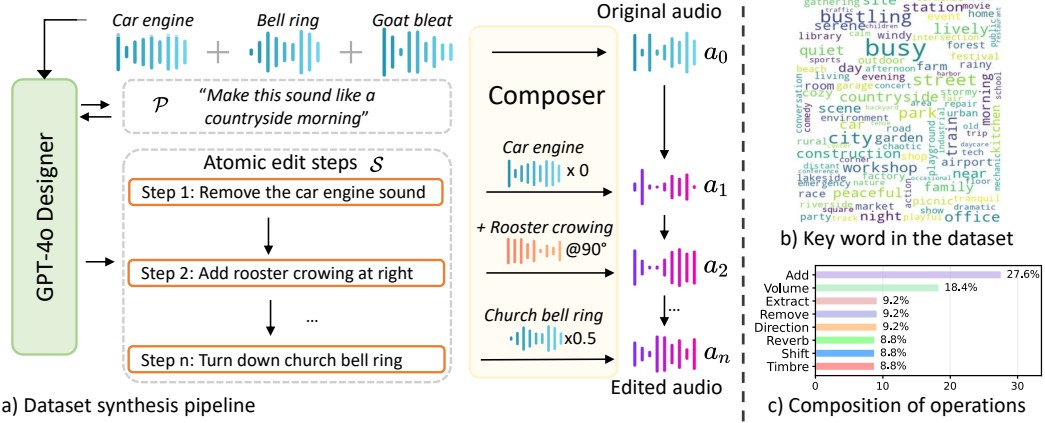

Figure 4: **Scalable data synthesis.** a) pipeline: we sample audio clips from databases with text labels and compose them into original audio $a_0$; These text labels are then fed into GPT-4o, which is prompted to design a declarative instruction $\mathcal{P}$ and generate corresponding atomic steps $\mathcal{S}$. We compose the target audio $a_1, a_2, ..., a_n$ following the atomic steps sequentially with rule-based composer. b) Key words in the declarative instruction. c) The proportion of each single-step edit operation in the dataset.

edits $\mathcal{S} = \{s_1, s_2, ..., s_n\}$, including both `add` operations and modifications to existing events (e.g., `remove`, `turn up/down`, `change sound direction`).

Once the sound designer has provided the sequential edit operations, an audio signal processing based composer takes over. To generate edit audio pairs, the composer first synthesizes the initial audio $a_0$ by superposing the $K$ audio clips. Spatial effects are rendered with direction-dependent phase and amplitude on two channels. For each atomic edit $s_i$, we proceed as follows:

- If $s_i$ modifies an existing event, we update its volume level or sound direction.
- If $s_i$ is an *add* operation, we retrieve a new clip from the database with a matching label.

Since each sampled audio event is *independently editable*, an edit step $s_i$ that modifies an existing event can be converted into event-level parameter adjustments, without altering any other sources in the mixture $a_{i-1}$. For example, to simulate *"remove the car engine sound"* (Fig. 4), we set the car engine's volume to zero to generate $a_1$. To simulate *"add rooster crowing at right"*, we retrieve a rooster clip, apply the specified spatial effect, and superpose it on $a_1$ to obtain $a_2$. To simulate *"turn down the dog bark"*, we reduce the volume of the corresponding clip. This allows us to generate a complete editing trajectory $a_1, a_2, ..., a_n$ by progressively updating event-level parameters and re-composing the audio scene. Our resulting dataset contains declarative editing instructions, atomic edit sequences, and the audios for the full editing trajectory. More details available in Appx. A.4.

## 4 EXPERIMENT

### 4.1 SETUP

**Dataset**. We use a combination of datasets including AudioCaps (Kim et al., 2019), VGGSound (Chen et al., 2020), FSD50k (Fonseca et al., 2021), ESC50 (Piczak, 2015), and WavCaps (Mei et al., 2024). We adopt dataset cleaning pipelines following prior work (Hai et al., 2024; Sun et al., 2024; Wang et al., 2023). Each audio is trimmed or padded into 10 seconds with a sampling rate of 24 K. We sample 2-5 audio events and use GPT-4o to create 240k training pairs and 2k evaluation pairs of declarative audio editing data (from test set of AudioCaps) to train ALM planner and evaluate the whole editing pipeline. We present the keyword in the declarative instructions in Fig. 4b. We also expand the size of single-step editing data pairs $(s_t, a_{t-1}, a_t)$ to 1M. Fig. 4c shows the proportion of each operation. This dataset is to train our audio editor. We use 3k single-step pairs from AudioCaps test set to evaluate the results. More details in Appx. A.1.

**Spatial Sound.** We use a interaural distance of 17 cm and adopts a cardioid directivity gain pattern to model the sound level differences between left/right ear when sound events are coming from different directions. To introduce room realism, we further use PyRoomAcoustics (Scheibler et al., 2018)

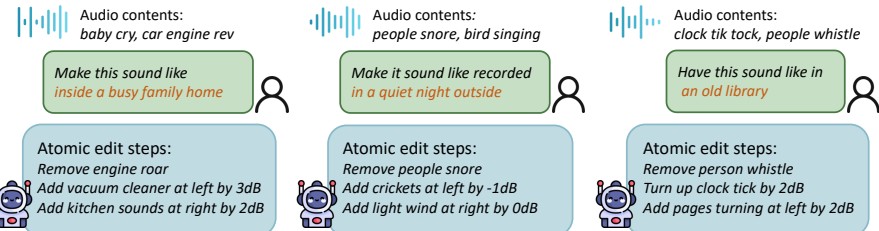

Figure 5: Examples of ALM's output detailed steps. Our ALM module identifies events in the original audio clips and reasons on the given declarative instruction to produce aligned editing steps.

to simulate room impulse response with reverberation time (RT60) (Sun et al., 2024) ranging from 0.3 s to 1.2 s. Room size, microphone orientation, and source/mic placement are randomized during synthesis. More details on acoustic simulation can be found in Appx. A.2.

**Metrics**. To evaluate edit quality and diversity, we use common metrics in audio generation and editing (Liu et al., 2023a; Wang et al., 2023), including Fréchet Distance (FD), Kullback-Leibler divergence (KL), Fréchet Audio Distance (FAD), Inception Score (IS), and Log-Spectral Distance (LSD). We use CLAP score to measure the semantic similarities between the edited audio and the text prompt. For spatial audio, we calculate GCC MSE (GCC) based on Generalized Cross-Correlation with Phase Transform (GCC-PHAT), and use StereoCRW (Chen et al., 2022) to produce stereo audio features to evaluate CRW MSE (CRW) and Fréchet Stereo Audio Distance (FSAD).

**Baselines**. To evaluate the declarative instruction based audio editing task, we first train an end-to-end version of Audit (Wang et al., 2023) that directly predicts the final-step edited audio $a_n$ conditioned on the original audio $a_o$ and declarative instruction $\mathcal{P}$ in one step without ALM. We extend the mono-channel Audit to our binaural setting, where we stack the left and right channels of the mel-spectrograms as the model inputs. We also evaluate various zero-shot and training-required editing methods based on the ALM's outputs to perform multi-step sequential editing. The zero-shot methods include SDEdit (Meng et al., 2022), DDIM Inversion (Mokady et al., 2022), ZETA (Manor & Michaeli, 2024), and AudioEditor (Jia et al., 2024). For a fair comparison, we replace these methods' generation backbone with the current SOTA methods. In AudioEditor, we replace Affusion with BEWO (Sun et al., 2024) to support binaural editing. In SDEdit, DDIM Inversion, and ZETA, we use Stable-Audio-Open (Evans et al., 2024b) as the backbone. For sequential editing with a trainable editor, we also use an Audit baseline trained on single-step audio editing. In addition, we evaluate the same set of baseline methods on single-step audio editing tasks. More details available in Appx. B.1.

**Implementation details.** SmartDJ ALM is initialized from AF2 (Ghosh et al., 2025a) with 3B parameters. During training, we freeze the AF-CLAP module and fine-tune the adapter layers and LLM with a LoRA module for 20 epochs with a batch size of 24. For LDM, it uses velocity prediction and CFG rescaling technique (Lin et al., 2024) to adjust the magnitude of the predicted velocity and avoid over-exposure. It is trained on single-step editing data with a batch size of 256 of 500k iterations. 10% text is replaced with empty strings to enable unconditional modeling. The learning rates for the ALM and LDM training are 1e-5 and 5e-5 with AdamW. All experiments are conducted with four NVIDIA L40S GPUs. More details available in Appx. B.2.

## 4.2 Results

**Declarative instruction audio editing.** We first show inference examples from our ALM module in Fig. 5. The ALM-generated atomic editing steps accurately align both the original audio contents and the declarative editing instructions. For example, it correctly removes audio event *engine roar* when transferring audio scene into a *busy family home* vibe. It also removes *people whistle* and adds *pages turning* to enhance the immersion of being in an *old library*. More examples in Appx. C.1.

We present the results of the declarative instruction editing task in Tab. 1. The first row is the end-to-end Audit baseline, which iss directly trained on declarative instructions and the final target audio, showing the worst performance overall. Since no prior method can interpret the declarative instructions, we use the same set of ALM-generated atomic steps to guide all audio editing models in the multi-step evaluation, including the baselines and our method. We compare the edited audios from each method with 1k reference audios. Our method achieves the lowest metric in FD, FAD, LSD, and comparably low in KL, indicating the smallest discrepancy from the reference audios. It

| Framework | Method | Training | Speed | FD ↓ | FAD ↓ | KL↓ | LSD↓ | CLAP↑ |
|---|---|---|---|---|---|---|---|---|
| w/o ALM | Audit | ✓ | 2.07s | 28.56 | 10.00 | 3.07 | 1.93 | 0.11 |
| w/ ALM | SDEdit | ✗ | 301s (74.6s) | 19.66 | 3.71 | 3.25 | 2.22 | 0.17 |
| | DDIM | ✗ | 331s (82.1s) | 24.70 | 9.43 | 4.06 | 2.20 | 0.07 |
| | ZETA | ✗ | 356s (88.2s) | 20.74 | 3.73 | 2.92 | 2.21 | 0.20 |
| | AE | ✗ | 406s (101s) | 19.91 | 4.99 | 3.21 | 2.08 | 0.19 |
| | Audit | ✓ | 11.6s (2.07s) | 21.50 | 5.67 | **2.80** | 1.49 | 0.18 |
| | SmartDJ (Ours) | ✓ | 13.1s (2.40s) | **10.60** | **1.52** | 2.84 | **1.40** | **0.21** |

Table 1: Quantitative results of the whole pipeline from declarative instructions to audio editing. Speed in (·) is the time for a single-step edit. AE denotes AudioEditor; DDIM denotes DDIM Inversion.

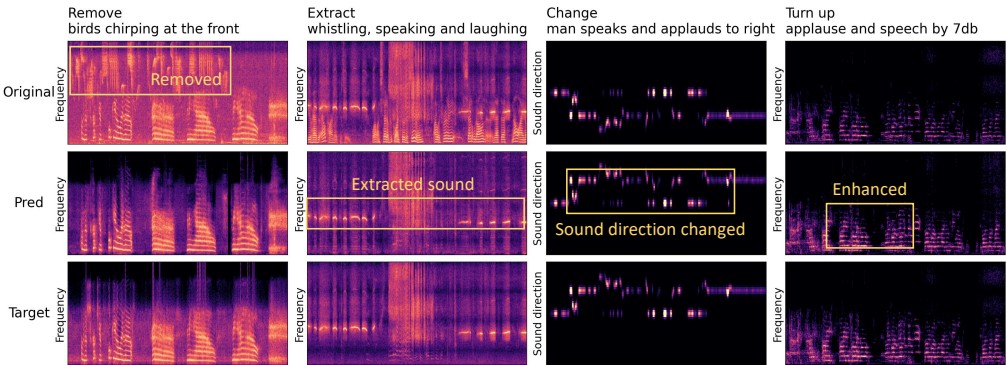

Figure 6: Examples on `Remove`, `Extract`, `Change`, `Turn up` operations. SmartDJ edited audio are closely aligned with the ground truth. Yellow boxes highlight edited sound components.

also delivers a high IS metric, showing strong audio quality and diversity. In addition, the highest CLAP score demonstrates the best semantic alignment between the edited audio and instruction.

**Inference time.** We also provide inference speed analysis in Tab. 1. The inference speed reported outside the brackets denote the total time to complete the entire declarative instruction edit, while the values inside indicate the per-round inference time of LDM. On average, SmartDJ's ALM requires 4.8s to generate one set of atomic editing instructions. With multi-round reasoning, SmartDJ completes a full edit in 13.1s, which is significantly faster than training-free baselines. Our approach is slower compared with end-to-end Audit (2.07s) or Audit with ALM (11.6s), but this trade-off yields substantially better editing quality and alignment with target instructions.

**Single-step audio editing.** We present the results of single-step audio editing operations. Tab. 2a shows results on `add`. Our method consistently outperforms baseline methods in the edited audio, including better similarities to the ideal target audio (lowest FD, FAD and KL), and higher quality and diversity shown by the highest IS. Furthermore, stronger spatial metrics also indicate that the stereo audio characteristics are preserved better by SmartDJ.

Tab. 2a also shows the performance on `Remove` and `Extract` tasks. The results clearly indicate that our method delivers the best performance in aligning with the ground truth edited audio across both operations. We show the results on `Volume`, `Time`, `Reverberation`, `Timbre` and `Change sound direction` tasks in Tab. 2b and 2c. Our method again shows stronger performance over Audit and training-free baselines, which demonstrates SmartDJ has better fine-grained manipulation in audio event properties. This is because Audit's VAE operates on mel-spectrograms and discards phase information, which is critical for spatial cues. SmartDJ employs a diffusion transformer, which provides stronger long-range temporal modeling and richer cross-attention conditioning. These architectural upgrades allow edits to be both semantically precise and spatially coherent. Some examples of spectrogram visualization are shown in Fig. 6. More qualitative comparisons can be found in Appx. C.3.

**Human evaluations.** We conducted a blind, pairwise A/B preference survey with 19 participants. Each participant evaluated 20 randomly sampled pairs (10 declarative editing, 10 single-step editing). Each data pair consists of the original audio, the editing prompt, SmartDJ edited result, and edited results from a random competing baseline. In each pairwise comparison, the participants were asked

| Method | Add | | | | | | | Remove/Extract | | | | | | |
|---|---|---|---|---|---|---|---|---|---|---|---|---|---|---|
| | FD↓ | FAD↓ | KL↓ | LSD↓ | GCC↓ | CRW↓ | FSAD↓ | FD↓ | FAD↓ | KL↓ | LSD↓ | GCC↓ | CRW↓ | FSAD↓ |
| SDEdit | 25.79 | 4.46 | 2.57 | 2.10 | 73.15 | 209.67 | 0.28 | 36.32 | 4.13 | 2.20 | 1.88 | 52.31 | 102.48 | 0.28 |
| DDIM | 28.84 | 7.14 | 2.62 | 2.11 | 66.97 | 185.25 | 0.07 | 42.01 | 5.06 | 2.54 | 1.84 | 52.35 | 89.89 | 0.18 |
| ZETA | 29.38 | 4.14 | 2.44 | 1.79 | 73.15 | 203.60 | 0.35 | 31.64 | 3.36 | 1.77 | 1.98 | 53.06 | 101.18 | 0.27 |
| AE | 23.84 | 4.16 | 2.11 | 1.92 | 68.15 | 232.64 | 0.37 | 36.71 | 3.25 | 2.24 | 2.00 | 53.79 | 107.48 | 0.36 |
| Audit | 27.82 | 5.11 | 1.94 | 1.48 | 74.37 | 217.49 | 0.21 | 42.48 | 6.73 | 1.96 | 1.64 | 62.06 | 132.72 | 0.67 |
| SmartDJ | **17.74** | **2.07** | **1.38** | **1.41** | **39.05** | **65.90** | **0.02** | **20.27** | **2.29** | **0.95** | **1.70** | **5.22** | **16.55** | **0.01** |

(a) Operation `add` and `remove/extract`

| Operation | Method | FD↓ | FAD↓ | KL↓ | LSD↓ | GCC↓ | CRW↓ | FSAD↓ |
|---|---|---|---|---|---|---|---|---|
| Volume | Audit | 36.66 | 4.56 | 1.33 | 1.04 | 83.71 | 121.10 | 0.53 |
| | SmartDJ | **9.19** | **0.82** | **0.24** | **1.01** | **1.75** | **20.30** | **0.01** |
| Time | Audit | 33.76 | 4.15 | 1.52 | 1.31 | 167.25 | 81.96 | 0.70 |
| | SmartDJ | **10.59** | **1.02** | **0.45** | **1.12** | **2.38** | **16.90** | **0.08** |
| Reverb | Audit | 39.56 | 4.36 | 2.20 | 1.09 | 157.89 | 164.60 | 0.69 |
| | SmartDJ | **10.83** | **0.89** | **0.55** | **1.04** | **15.20** | **39.00** | **0.09** |
| Timbre | Audit | 27.11 | 2.67 | 1.02 | 1.10 | 60.21 | 164.60 | 0.61 |
| | SmartDJ | **8.80** | **0.71** | **0.33** | **1.05** | **3.04** | **29.90** | **0.09** |

(b) Operation `Volume`, `Time`, `Reverb` and `Timbre`

| Method | Change Sound Direction | | | | | | |
|---|---|---|---|---|---|---|---|
| | FD↓ | FAD↓ | KL↓ | LSD↓ | GCC↓ | CRW↓ | FSAD↓ |
| SDEdit | 29.69 | 2.85 | 1.37 | 1.31 | 70.18 | 212.78 | 0.25 |
| DDIM | 41.52 | 57.56 | 2.19 | 1.20 | 82.09 | 211.98 | 0.09 |
| ZETA | 28.35 | 2.49 | 1.06 | 1.42 | 67.29 | 218.11 | 0.22 |
| AE | 33.29 | 3.69 | 1.42 | 1.39 | 40.28 | 167.05 | 0.16 |
| Audit | 30.43 | 3.65 | 1.03 | 0.94 | 72.17 | 205.18 | 0.51 |
| SmartDJ | **9.45** | **0.67** | **0.27** | **1.02** | **26.02** | **48.69** | **0.01** |

(c) Operation `change sound direction`

Table 2: Quantitative results on all individual audio editing operations.

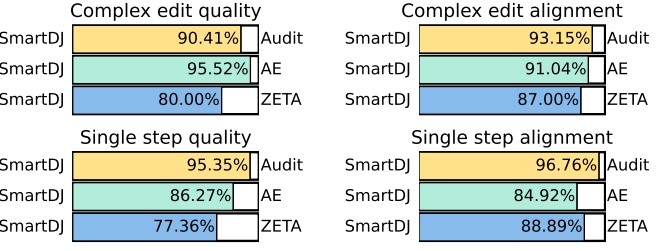

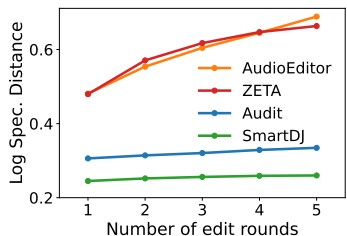

Figure 7: User study results. In both audio quality and text/audio alignment, SmartDJ is consistently preferred over baselines in the declarative instruction editing task and single-step tasks.

Figure 8: Similarity with original audio after multiple editing rounds.

to select one of the edited audio with higher quality, better alignment to the text or spatial instructions, and better alignment to the original audio. We separate the evaluation for declarative instruction audio editing and single-step editing in Fig. 7. In both tasks, SmartDJ is much preferred over all competing methods with a higher win-rate. Our method delivers the best audio quality and the best alignment with both declarative instructions and single-step instructions. We also use a two-sided Z-test and find SmartDJ demonstrates statistical significance ($p < 0.001$). We find that our 95% confidence intervals further confirm robustness, with all lower bounds (ranging from 66.1% to 94.2%) exceeding the random chance threshold of 50%. More details of the user study are listed in Appx. C.4.

### 4.3 ABLATION STUDIES

**Performance of ALM generate editing steps.** We use BERTScore (Zhang et al., 2019) to evaluate the ALM generated results. It measures semantic similarity between two sentences with a pretrained BERT model. For our task, each detailed editing plan consists of multiple atomic steps. To ensure fair evaluation, we group atomic operations by type (add, remove and so on), and compare only between matching operation types in the GT and generated plan. We then compute BERTScore for each operation type and report the average across all operation groups (results in 91.8% Precision, 92.0% Recall and 91.5% of F1). We also provide some failure case analysis in Appx. C.2, where the ALM is not able to fully understand contradictory instructions.

**Audio quality over multi-round editing.** Since the declarative instruction audio editing task involves a sequence of editing, unchanged content must remain intact after multiple steps. To evaluate this, we design a "round-trip" edit experiment: we perform the operations "add the sound of $\mathcal{A}$" and "remove the sound of $\mathcal{A}$" on an audio clip for five rounds, where $\mathcal{A}$ is a pseudo audio label. For an ideal audio editor, this sequence of round-trip operation should exactly reconstruct the original audio. We measure the LSD between each round's edited output with the original audio clip (Fig. 8). SmartDJ consistently achieves the lowest LSD, indicating the smallest drift from the original content. This demonstrates that our method preserves the unedited audio content under repeated editing actions.

| Study Objectives | Variation | FD↓ | FAD↓ | KL↓ | LSD↓ | CLAP↑ |
|---|---|---|---|---|---|---|
| ALM module | w/o ALM | 23.6 | 3.14 | 2.91 | 1.84 | 0.137 |
| | Caption+GPT-4o+SmartDJ editor | 16.8 | 2.70 | 2.96 | 1.45 | 0.184 |
| | w/ SmartDJ ALM | **14.7** | **1.53** | **2.85** | **1.42** | **0.238** |
| Editing order | Add → Modify → Remove | 14.9 | 1.59 | 2.88 | 1.48 | 0.234 |
| | Random order | **14.7** | 1.56 | 2.88 | 1.47 | 0.237 |
| | Remove → Modify → Add | **14.7** | **1.53** | **2.85** | **1.42** | **0.238** |
| Extract operation | AudioSep (Liu et al., 2024b) | 27.1 | 2.86 | 0.88 | **1.63** | N/A |
| | SmartDJ | **25.7** | **2.55** | **0.78** | 1.71 | N/A |

Table 3: Model ablations.

**Effectiveness of ALM.** We conduct an ablation by removing the ALM module and training a variant of the LDM end-to-end. As shown in Tab. 3, SmartDJ performs significantly worse without ALM, showing the importance of ALM's intermediate reasoning capabilities. ALM enables the model to produce semantically coherent edits aligned with the declarative instruction and the original audio.

**Choice of ALM.** To assess the sensitivity of SmartDJ to the choice of ALM backbone, we replaced our AF2-based ALM with LTU (Gong et al., 2023) while keeping the editor unchanged. As shown in Tab. 3, using LTU results in slightly worse performance. We attribute this gap primarily to the fact that our ALM is pretrained on a much larger and more diverse corpus, giving it stronger performance on decomposing instructions. Nevertheless, the LTU+SmartDJ editor variant still produces reasonable results. This indicates that our framework is generalizable to different model ALM architectures.

**Can Caption Model + LLM replace ALM?** We evaluate a captioning-based alternative pipeline, using a DCASE captioning model (Xu, 2024) to describe the audio and GPT-4o to produce step-wise instructions executed by SmartDJ editor. As shown in Tab. 3, this captioning-based pipeline performs consistently worse than SmartDJ. We attribute the performance drop to two factors: (1) Captioning models summarize only dominant events and often miss fine-grained or overlapping sounds. (2) Missing or misidentified events in the caption propagate through the LLM's reasoning, compounding into hallucinated or non-executable editing instructions with cumulative errors. But future works can explore using a strong audio captioning model to directly get the audio events and prompt LLM to achieve the declarative editing.

**Editing order.** We adopt a simple ordering strategy of remove → modify (volume/direction) → add, which intuitively avoids removing newly inserted content. To test the impact of ordering, we also experiment with two alternatives: randomized and reversed order. As shown in Tab. 3, both alternatives result in only marginal degradation compared to our default ordering. This indicates that the editing order has minimal influence on the final results, suggesting that ALM-generated instructions rarely contain conflicting operations.

**Compare with sound separation model.** We compare SmartDJ with a recent sound separation model AudioSep (Liu et al., 2024b) on the Extract operation. Ours achieves comparable or slightly better performance (Tab. 3). This suggests that while our model is designed for general-purpose audio editing, it is competitive on target sound separation tasks.

## 5 DISCUSSION

This work takes an initial step toward declarative stereo audio editing by bridging high-level instruction understanding with low-level generative transformation. Rather than treating audio editing as a monolithic operation, our framework decomposes complex user intent into atomic and sequentially executable steps. While extending the generative audio editor to support entirely new task-specific operations currently requires retraining, many realistic edits can already be composed from a small set of reusable atomic operations. Looking forward, a promising avenue is to unify reasoning and generation through end-to-end joint training of audio language models and generative backbones, enabling tighter coupling between semantic intent and low level signal editing. More broadly, we believe this paradigm opens new opportunities for interactive audio creation systems and pushes toward more general-purpose audio intelligence.

# 6 ETHICS STATEMENT

Our work focuses on audio editing for research and creative applications such as immersive media, conferencing, and sound design. The dataset is generated from publicly available sound event libraries and synthetic mixing, without personal or sensitive recordings. We encourage responsible use aligned with academic and creative purposes.

# 7 REPRODUCIBILITY STATEMENT

We provide detailed descriptions of the model architectures, training objectives, and data generation pipeline in the main paper and appendix. Hyperparameters, training configurations, and dataset construction details are included to ensure reproducibility. Code, pretrained models, and the synthesized dataset will be released upon acceptance to facilitate replication of our results.

## ACKNOWLEDGMENTS

We thank the members of the WAVES Lab at the University of Pennsylvania for their valuable feedback. We are grateful to the anonymous reviewers for their insightful comments and suggestions.

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

# A  DATASET CURATION PROCESS

## A.1  DATASET PREPARATION

We construct our training corpus by merging several publicly-available audio dataset, including VGG-Sound, AudioCaps, WavCaps, ESC-50, and FSD50K. Since some of these sets provide audio captions rather than discrete audio label, we first convert every caption to audio labels with GPT-4o-mini API. We only retain single-labeled audio clip and any clip whose caption maps to multiple events is discarded. A CLAP model scores the semantic correspondence between the audio and its new label. Samples with a similarity score below 0.3 are filtered out. The remaining clips from all sources are finally mixed into one large dataset that we use for subsequent data curation.

## A.2  DETAILED IMPLEMENTATION OF SPATIAL AUDIO

We simulate room impulse responses using PyRoomAcoustics (Scheibler et al., 2018) to generate stereo audio in diverse environments. Rectangular rooms are created with horizontal dimensions sampled from

$$R = [R_0, R_1, R_2], \qquad R_0, R_1 \sim U(4, 10)\,\mathrm{m}, \quad R_2 = 3\,\mathrm{m}, \tag{6}$$

All microphone and source positions are sampled uniformly inside the room with a minimum $0.2$m to each wall, and the microphone azimuth orientation is drawn at random.

The reverberation time of RIR is sampled as $\mathrm{RT60} \sim U(0.3, 1.2)\,\mathrm{s}$, covering different level of reverberation conditions.

**Directional sampling.**  We consider semantic source directions (left, front-left, front, front-right, right) and map each label to an azimuth angle $\theta \in \{180°, 135°, 90°, 45°, 0°\}$. To introduce variability, we perturb the azimuth with Gaussian noise (standard derivation of $5°$). This $\theta$ represents the relative angle between the microphone orientation and the sound source.

**Binaural rendering with cardioid gain.**  We use a two-microphone array with spacing $d = 17$cm to simulate human ears. Let $x(t)$ denotes the dry source signal, and let $\alpha$ be the incident angles of the source relative to the left or right microphone normals. We use a cardioid gain pattern $G(\alpha) = \frac{1}{2}\big(1 + \cos\alpha\big)$. The rendered left and right ear signals are

$$y_L(t) = x(t) * h_L(t), \tag{7}$$
$$y_R(t) = x(t) * h_R(t), \tag{8}$$

where $h_L(t)$ and $h_R(t)$ are the RIR synthesized at the left and right ear position, $*$ denotes convolution. The synthesized RIR considers the directional dependent gain pattern when combining the acoustic ray coming from different directions. As a result, the interaural time and level difference are encoded within the synthesized RIR to create spatial audio experience.

## A.3  ATOMIC EDIT ACTIONS

We explain the details on creating the single-step atomic edit data pairs. For this single-step audio editing, we have an original audio $a_{i-1}$, a single atomic edit operation $s_i$. Base on atomic edit operation $s_i$, we can generate the edited audio $a_i$.

***Add.***  Assume the original audio is a mix of audio content $A+B+C$. To *add* a new content into this original audio, we sample a new audio content D from the database and mix the D with the original audio $A+B+C+D$. The atomic template is *"Add the sound of {dog barking} at {right} with {3} db"*. The contents inside the {} can be changed to other sound events or sound attributions. We support various sound direction (left, front and right) and dynamic volume adjustment.

***Remove.***  Given an original mix $A+B+C$, let $B$ be the undesired source. We suppress $B$ so the output becomes $A+C$. The atomic template is Atomic template: *"Remove the sound of {bird chirping} {at right}"*. The directional phrase in braces is optional. If there are similar audio contents in the same clip, the spatial features enables to manipulate it precisely.

**Extract.** Starting from the same mix $A+B+C$, we isolate one target source $A$ and mute everything else, yielding only $A$. The atomic template is *"Extract the sound of {speaking} {at the right}"*. The direction is optional.

**Turn up/down** To change loudness of a specific source $B$, we scale it by $\alpha$, where $\alpha > 1$ is for "up" and $0 < \alpha < 1$ is for "down". The resulted audio clip is $A + \alpha B + C$. The atomic template is *"Turn {up / down} the sound of {engine rev} by {2} dB"*. We also support a dynamic range of volume adjustment.

**Change sound direction.** We alter only the spatial cues of a specific source $C$, producing an edited version $C'$ while leaving the other tracks untouched: $A+B+C'$. The atomic template used is *"Change the sound of {baby crying} {from front} to {right}"*. The "from" clause could also be omitted.

**Time shift.** Given an original mix $A+B+C$, let $B$ be the target source. We apply time shifting to $B$ so the output becomes $A+B'+C$. The atomic template is Atomic template: *"Shift time of the sound of {bird chirping} {by 3 seconds}"*.

**Add reverberations.** Given a mix $A+B+C$, we apply reverberation to a specific target source (e.g., $B$) by convolving it with a room impulse response (RIR). The resulting edited clip becomes $A + B' + C, \quad B' = B * \text{RIR}_{\text{level}}$. We support three reverberation levels, each corresponding to an RIR with a different decay time (RT60):

- Low reverberation: RT60 $\approx 0.4$ s
- Mid reverberation: RT60 $\approx 0.8$ s
- High reverberation: RT60 $\approx 1.2$ s

The atomic template is *"Add reverberation to the sound of {violin} {at left} of {low / mid / high} level"*. A directional qualifier (e.g., "at left") is optional. The reverberation is applied only to the selected source, while all other sources remain unchanged.

**Timbre adjustment.** We modify the spectral characteristics of a specific source $C$ while preserving its spatial cues and temporal structure. The resulting signal $C'$ is obtained via frequency-domain filtering, and the output becomes $A+B+C'$. We support several interpretable timbre transformations commonly used in audio production:

- Bright: boost high frequencies $> 3$ kHz
- Dark: attenuate high frequencies $> 3$ kHz
- Warm: boost mid–high components (300 Hz–4 kHz)
- Cold: boost $> 6$ kHz and attenuate $< 300$ Hz
- Muffled: attenuate frequencies $> 1.5$ kHz

The atomic template is *"Change the timbre of the sound of {acoustic guitar} to {bright / dark / warm / cold / muffled}"*. These adjustments operate at the source level, allowing precise sound-color manipulation even when multiple acoustically similar sources appear within the same audio clip.

## A.4 Complex audio editing dataset curation

In the dataset curation process, we first sample 2-4 audio labels in the database, with LLM randomly assigned volume and sound directions. We then call the GPT-4o batch API with sound sources and attributions. For each API call, we provide 15 sets of sound sources. Through API call, each set of sound sources will return a single data pair containing Declarative editing instruction and the corresponding atomic editing steps. We follow these atomic editing steps to manually generate the step-by-step target edited audio with the rules in A.3. We generate 240K data pairs for complex audio editing for training. We generate 2K data pairs from AudioCaps test/validation set for evaluation.

For the single step audio editing pairs $(s_i, a_{i-1}, a_i)$, we further scale up the this dataset size to 1M to train the LDM audio editor and we also generate another 2K extra audio data pairs to evaluate the performance of single step audio editing. This scaled-up single step audio editing dataset keeps some instructions with audio captions, improving the audio editor's robustness to different audio contents in the atomic edit instructions. Tab. 4 shows a comparison between current audio generation and editing dataset, where ImmerseDiffusion (Heydari et al., 2025) and YT-Ambigen (Kim et al., 2025) are two current spatial audio generation work.

| Dataset | Task | Hours | Number | Data Type | Format |
|---|---|---|---|---|---|
| **Audio Generation Datasets** | | | | | |
| ImmerseDiffusion | Text to Audio | 6000 | 1.1M | Text + Audio | FOA |
| YT-Ambigen | Video to Audio | 142 | 102K | Video + Audio | FOA |
| BEWO-1M (text) | Text to Audio | 2800 | 1M | Text + Audio | Binaural |
| BEWO-1M (image) | Image to Audio | 54 | 2.3K | Image + Audio | Binaural |
| **Audio Editing Datasets** | | | | | |
| Audit | | 1605 | 578K | Text + Audio | Mono |
| | Add | 197 | 71k | | |
| | Remove | 197 | 71k | | |
| | Replacement | 139 | 50k | | |
| | Inpainting | 536 | 193k | | |
| | Super-resolution | 536 | 193k | | |
| SmartDJ Single-Step Editing | | 2778 | 1M | Text + Audio | Binaural |
| | Add | 768 | 277K | | |
| | Remove | 255 | 92K | | |
| | Extract | 256 | 92K | | |
| | Volume adjust | 510 | 183K | | |
| | Direction adjust | 255 | 92K | | |
| | Timing adjust | 245 | 88k | | |
| | Add Reverb. | 245 | 88K | | |
| | Timbre adjust | 244 | 88k | | |
| SmartDJ Declarative Editing | Declarative editing | 667 | 240K | Text + Audio | Binaural |

Table 4: Summary of current audio generation and editing datasets.

We provide the details of our *Base Prompt* for dataset curation as follow:

---

**Prompt**

You are an expert in spatial audio editing and sound design.

Your task is to generate complex audio editing instructions based on a given list of sound sources (labels). The sound sources will be provided as a list of full sentences (as strings), not character lists. Treat each sentence as a single atomic sound unit. Do not tokenize or split the sound sources into characters.

For example, if you are given: "a baby crying and a man talking; a bird is chirping; dog barking". You should consider "a baby crying and a man talking" as a complete sound source. "a bird is chirping" is another complete sound source and "dog barking" is another sound source. You then generate the step-by-step audio editing instructions based on the given complex instructions.

Task: you need to first brainstorm a complex audio editing instruction

- Imagine a realistic and creative soundscape editing for the given audios.
- You are not limited to the provided audio contents for the editing instruction. And the complex editing instructions should be brief.
- The complex editing instructions could be a soundscape transformation. For example:
  - - Make this sound like it was recorded in a bookstore
  - - Make this sound like a busy coffee shop
  - - Make this sound like a train station
  - - Make this sound like a forest at night
  - - Make this sound like a beach
  - - Make this sound like a sunny day
  - - Craft this sound to feel like a park
  - - Make this audio sound like a quiet farm
  - - Make this audio sound like a firework show
- Your generated complex instructions can be broader than these provided examples. Use your imaginations!

---

- But remember to keep them brief, the complex editing instructions ideally should not contain the actual sound sources.
- Then, based on the given sound event, give me the detailed editing instructions.

You then generate detailed editing instructions based on the complex audio editing instruction.

- Ensure the editing makes sense (e.g., no waves in a desert, no sheep indoors, rustling leaves in the forest, seas waves in the beach, no raining in the sunny day).
- You are encouraged to remove the original audio contents, but you are required to maintain at least one sound source, not removing all of them.
- Do not split or partially reference a sound source when applying operations.
- Use a combination of simple operations (but NOT necessarily all of them). For example:
  - - Add (e.g., thunderstorm, cat meowing) (for the add operation, you should add up to two additional sources that best align with the complex editing instructions)
  - - Remove (For remove, you should remove at least one sound sources (up to two sound sources), but you must keep at least one sound source!)
  - - Turn up/down (e.g., turn up/turn down the sound of xxx by xxx db (between 0 and 6db))
  - - Change sound direction.
  - - Add reveberations (low, medium, or high level)
  - - Change timbre (choose from bright, dark, warm, cold, muffled)
- Each "target" in remove/turn up/turn down/change must exactly match one of the provided "sound sources"
- For the "add" operation:
  - - The "target" must clearly describe the new sound being added (e.g., "crowd chatter", "rain", "footsteps on gravel").
  - - The "effect" should specify the volume and direction (e.g., "at front by 4dB").
  - - Do NOT use "none", "null", or placeholder values as the target. The target must always be a descriptive label of the added sound.
  - - The added sound should not duplicate any original sound source.
- The same operation can be repeated for multiple targets.
- When doing add, you can also have volume and direction attributes
- When editing existing sound sources, you can also have mixed attributes in terms of the volume, sound directions and timbre, also adding reverberations.
- But limit the total number of detailed editing steps to be between 2 to 5.
- You must ensure each step logically contributes to the final transformation.

Return the output in the following structured JSON format:

{

    "sound sources": ["...", "..."], here you should put the sound sources you are given

    "complex editing instruction": "...",

    "atomic editing steps": [

        {"operation": "add", "target": "...", "effect": "at xxx(left, front, right) by xxxdB"},

        {"operation": "remove", "target": "...", "effect": "None"},

        {"operation": "turn up/turn down", "target": "...", "effect": "xxxdB"},

        {"operation": "change", "target": "...", "effect": "to xxx (left, right, or front)"},

        {"operation": "add reverb", "target": "...", "effect": "(low, mid, or high)"},

        {"operation": "change timbre", "target": "...", "effect": "to xxx (bright, dark, warm, cold, muffled)"}

    ]

}

Do NOT break down sound sources into individual words or characters. Sound sources are complete strings and must remain so in the JSON.

# B IMPLEMENTATION DETAILS

## B.1 BASELINES IMPLEMENTATION

We present the following baseline methods to evaluate the complex audio editing task. One of the baselines is an end-to-end version of Audit without using ALM. All other baseline methods perform multi-step sequential editing with ALM's atomic editing step outputs.

**End-to-End Audit.** We first train an end-to-end version of Audit that directly predicts the final-step edited audio $a_n$ conditioned on the original audio $a_0$ and Declarative instruction $\mathcal{P}$ in one step without ALM. We extend the mono-channel Audit to our binaural setting, where we stack the left and right channels of the mel-spectrograms as the model inputs. Follow the original implementation, we convert 10s of audio into mel-spectrograms with a size of $80 \times 624$. using a hop size of 256, a window size of 1024, and mel-bins of size 80. We use the same model configurations in the original paper.

**SDEdit.** SDEdit is a zero-shot method that does not require training a new audio editing model. It uses an off-the-shelf text-to-audio (TTA) generation model, which we use Stable-Audio-Open, as it supports binaural audio generation. We use the default 200 total diffusion steps and start the reverse process from a timestep of 100. We use a classifier-free-guidance (CFG) scale of 7.5. The target caption is composed by concatenating the individual event captions after each editing step.

**DDIM Inversion.** Similar to SDEdit, we also use Stable-Audio-Open as the TTA generation model. We use the default 200 total diffusion steps and start the reverse process from a timestep of 100. We use a CFG scale of 7.5 for both the target and the source. The source text prompt and the target text prompt are composed by concatenating the individual event captions before and after each editing step, respectively.

**ZETA.** Similar to SDEdit and DDIM Inversion, we also use Stable-Audio-Open as the TTA generation model. We use the default 200 total diffusion steps and start the reverse process from a timestep of 100. We use a CFG scale of 7.5 for the target and a CFG scale of 1 for the source. The source text prompt and the target text prompt are composed by concatenating the individual event captions before and after each editing step, respectively.

**AudioEditor.** Specifically in AudioEditor, we replace Affusion with the spatial generator SpatialSonic in BEWO to support binaural editing. We use the default 100 total editing steps with 5 iterations per step to update text embedding for null-text inversion. For the addition task, we use the default punishment ratio (alpha) of -0.001. For the remove or extract task, we use the default punishment ratio (alpha) of 1.

**Audit with ALM.** To evaluate sequential editing with a trainable editor, we use an Audit baseline trained on single-step audio editing data with input audio $a_{i-1}$, atomic instruction $s_i$, and output edited audio $a_i$. The model and data configurations are the same with the end-to-end Audit baseline variant.

## B.2 SMARTDJ IMPLEMENTATION

**ALM.** Our ALM module is initialized from Audio Flamingo 2. This model contains an AF-CLAP audio encoder module to encode the mono-channel audio. The input 10-second audio is first resampled to 16KHz and transformed into a dense audio features $z_a \in \mathbb{R}^{64 \times 2048}$. The mono-channel CLAP encoder understands the audio semantics and sound events, which is enough to reason the atomic steps to make the edited audio semantically align with the Declarative instruction. This audio encoder is followed by a representation transformation layers that expand the model capacity. This module has three self-attention layers to the audio feature representation, each with 8 heads and a dimension of 2048. Following this, gated cross-attention layers are used to condition audio representations on the LLM. The LLM uses Qwen2.5-3B, a decoder-only causal LLM with 3B parameters, 36 hidden layers, and 16 attention heads. During training, we keep both the AF-CLAP and LLM frozen during training. The audio representation transformation layers are fully optimized. We apply LoRA only to the gated cross-attention layers with a rank of 16.

**DiT architecture** We conduct experiments at 24KHz audio sample rate. The waveform latent representation operates at 50Hz and consists of 128 channels. Our DiT model has 24 blocks, each

with a channel of 1024, 16 attention heads, and an FFN (Feed-Forward Network) dimension of 4096, resulting in total of 597M parameters. During training, we begin with a 50K linear warm-up phase and the result is training with a cosine learning rate decay. In the inference, we use 100 diffusion steps and a guidance scale of 4, with guidance rescale of 0.8.

## C   MORE RESULTS

In this section, we provide more qualitative results on the ALM-inferenced editing steps and the LDM edited audio results.

### C.1   RESULTS ON ALM-INFERENCED ATOMIC EDITING STEPS

We show multiple examples on ALM-inferenced atomic editing steps. In each example, the "audio labels" are the ground truth audio labels for reference and they are not provided to the ALM module. Only the corresponding audios are provided to the ALM. We then input the declarative instruction to the ALM and it output the atomic editing steps.

---

**Example 1: Make this sound like a quiet afternoon in a garden**

Audio labels: clock tick, bird chirp, wind
Declarative instructions: Make this sound like a quiet afternoon in a garden
Atomic editing steps:
- Remove the sound of clock tick
- Turn up the the sound of bird tweet by 3dB
- Add the sound of gentle breeze at front by 2dB

---

**Example 2: Make this sound like a busy city street**

Audio labels: engine rev, church bell ring
Declarative instructions: Make this sound like a busy city street
Atomic editing steps:
- Remove the sound of bell ring
- Turn down the sound of engine rev by 2dB
- Add the sound of distant siren at left by 2dB
- Add the sound of traffic noise at front by 3dB

---

**Example 3: Make this sound like a bustling park on a sunny day.**

Audio labels: children scream, insect buzz, bird call, chainsaw run
Declarative instructions: Make this sound like a bustling park on a sunny day.
Atomic editing steps:
- Remove the sound of chainsaw run
- Add the sound of laughter at left by 3dB
- Turn down children scream by 2dB
- Change the sound of bird call to front

---

**Example 4: Make this sound like a serene beach**

Audio labels: water waves, whistling, music playing
Declarative instructions: Make this sound like a serene beach
Atomic editing steps:
- Remove the sound of whistling
- Turn up the sound of wave crash by 4dB
- Add the sound of seagulls calling at front by 3dB

---

---

**Example 5: Make this sound like a busy office**

Audio labels: typewriter tap, power tool drill
Declarative instructions: Make this sound like a busy office
Atomic editing steps:
- Remove the sound of drilling
- Turn up the sound of typewriter type by 2dB
- Add the sound of phone ringing at right by 3dB

---

**Example 6: Make this sound like a military base**

Audio labels: fire truck siren, gunshot, vehicle move
Declarative instructions: Make this sound like a military base
Atomic editing steps:
- Remove the sound of fire truck siren
- Turn up the sound of gunshot by 3dB
- Add the sound of helicopter blades at right by 2dB

---

**Example 7: Make this sound like a calm night in the countryside**

Audio labels: snoring, machine run, wind blow
Declarative instructions: Make this sound like a calm night in the countryside
Atomic editing steps:
- Remove the sound of machine run
- Turn down the sound of wind blow by 2dB
- Add the sound of owl hooting at right by 2dB

---

**Example 8: Make this sound like a bustling urban street**

Audio labels: horn honk, cat, siren blare, male laugh
Declarative instructions: Make this sound like a bustling urban street
Atomic editing steps:
- Remove the sound of cat
- Turn down the sound of siren blare by 2dB
- Add the sound of people chatter at front by 3dB
- Add the sound of traffic noise at left by 2dB

## C.2    FAILURE CASE ON ALM-INFERENCED ATOMIC EDITING STEPS.

To better illustrate how the ALM behaves under abstract or conflicting prompts such as "make it sound eerie yet cheerful," we conducted a test using this instruction across different audio inputs:

---

**Example 1: Make this sound like eerie yet cheerful**

Audio labels: child cries, wind blows
Declarative instructions: Make this sound like eerie yet cheerful
Atomic editing steps:
- Add the sound of laughing at right by 3dB
- Add the sound of person whispers at left by 2dB
- Turn up child cries by 2dB

---

**Example 2: Make this sound like eerie yet cheerful**

Audio labels: female speech, bell ringing
Declarative instructions: Make this sound like eerie yet cheerful
Atomic editing steps:
- Turn up the sound of female speech by 2dB

---

> - Add the sound of owl call at front by 3dB

Overall, these examples show that the ALM can reliably handle atmosphere-related cues (e.g., "eerie", "cheerful"), but it is still a challenging task when emotional or stylistic attributes conflict or lack clear instruction. This reflects an inherent limitation of mapping abstract emotions to concrete operations. Extending the ALM to better model emotional and stylistic intent is a natural direction for future work.

## C.3 RESULTS ON ATOMIC EDITING STEPS

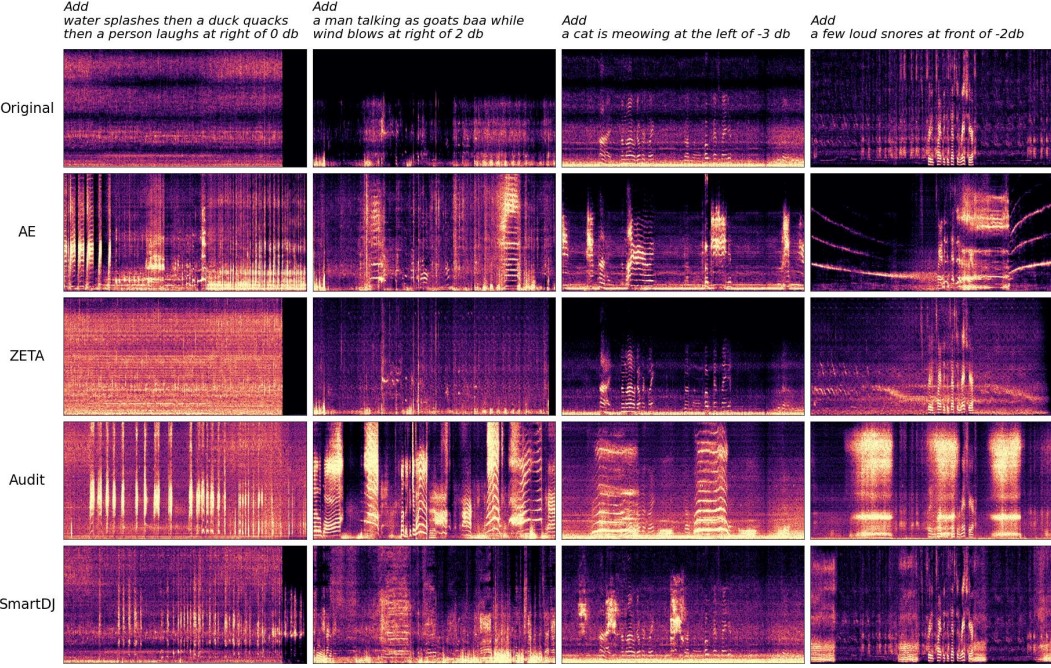

Figure 9: Examples on *add* operation. The top row is the original audio and the rest rows are the edited results. Only SmartDJ can keep the original audio clips while adding new audio content.

We provide more atomic editing results on *add* in Fig 9. As shown by the comparison with the original audio and the edited one, while the baseline method tends to replace the original clips and completely generate a new audio clip, SmartDJ can successfully keep the original contents and *add* new sound events into it.

We show more editing examples on *remove* and *extract* in Fig 10 and 12. Compared with baseline method, SmartDJ can effectively either remove unwanted or extract wanted audio contents. The edited results show good alignment with the ground truth edited audios.

Figure 11 presents additional qualitative results for the *change sound direction* task. The $Y$ axis in each figure is the sound direction heatmap. SmartDJ consistently relocates the source to the requested spatial direction, and its outputs align well with the ground truth edited audios. By contrast, Audit can not alter spatial effect, showing the limitations of using spectrogram audio encoder for direction-aware editing.

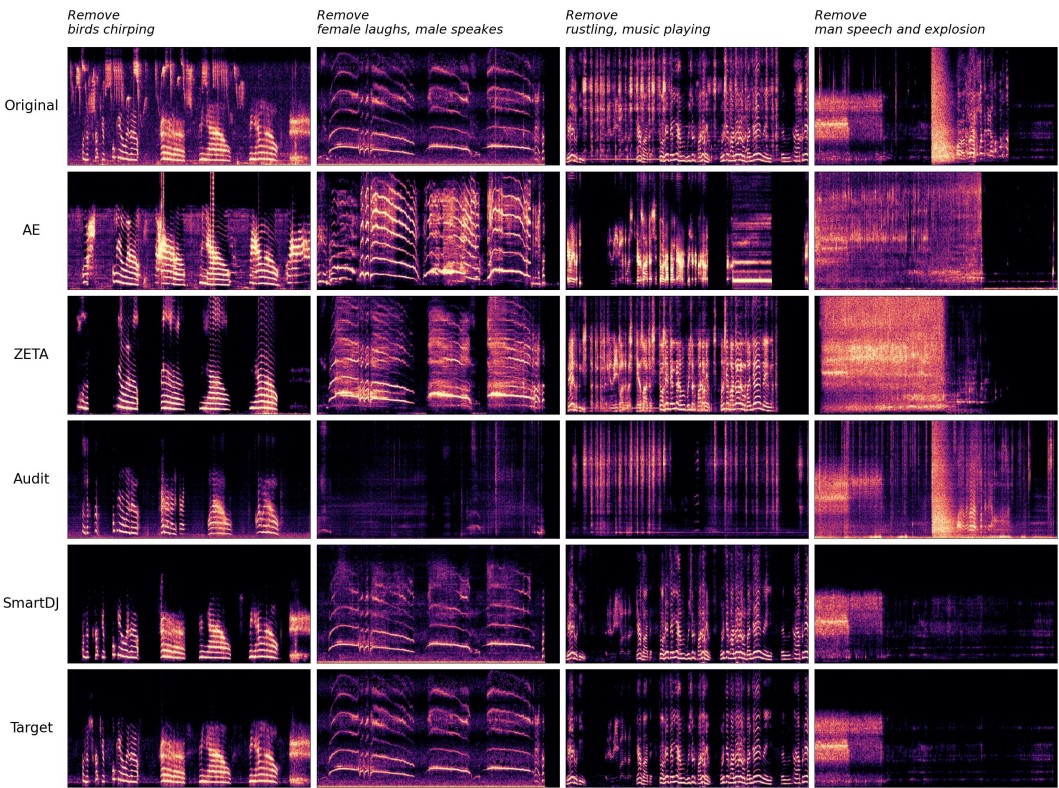

Figure 10: Examples on *remove* operation. The top row is the original audio, and the bottom row is the target audio. Only SmartDJ can completely remove unwanted audio parts and keep the remaining part unchanged.

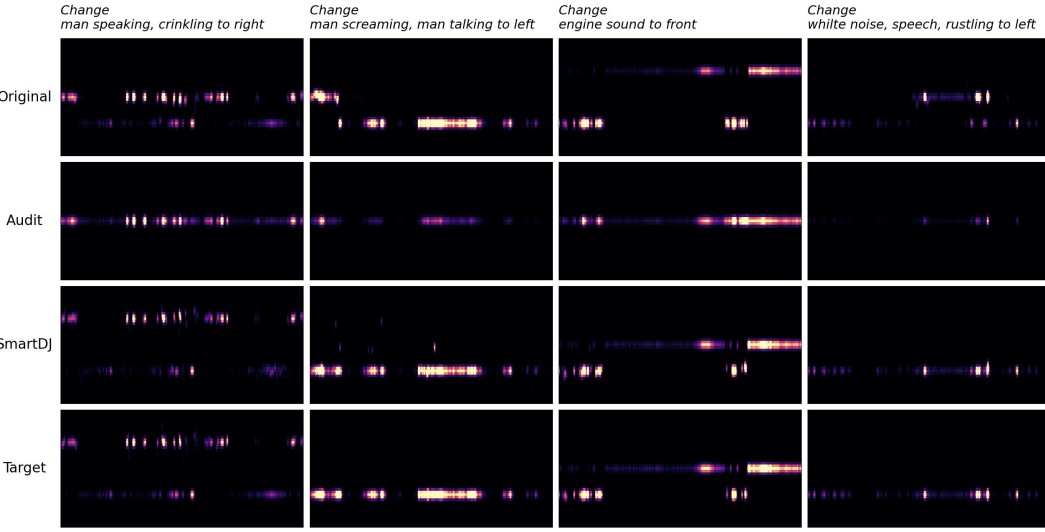

Figure 11: Examples on *change sound direction* operation. The top row is the original audio, and the bottom row is the target audio. Only SmartDJ can perfectly edit the sound directions that match closely with the target.

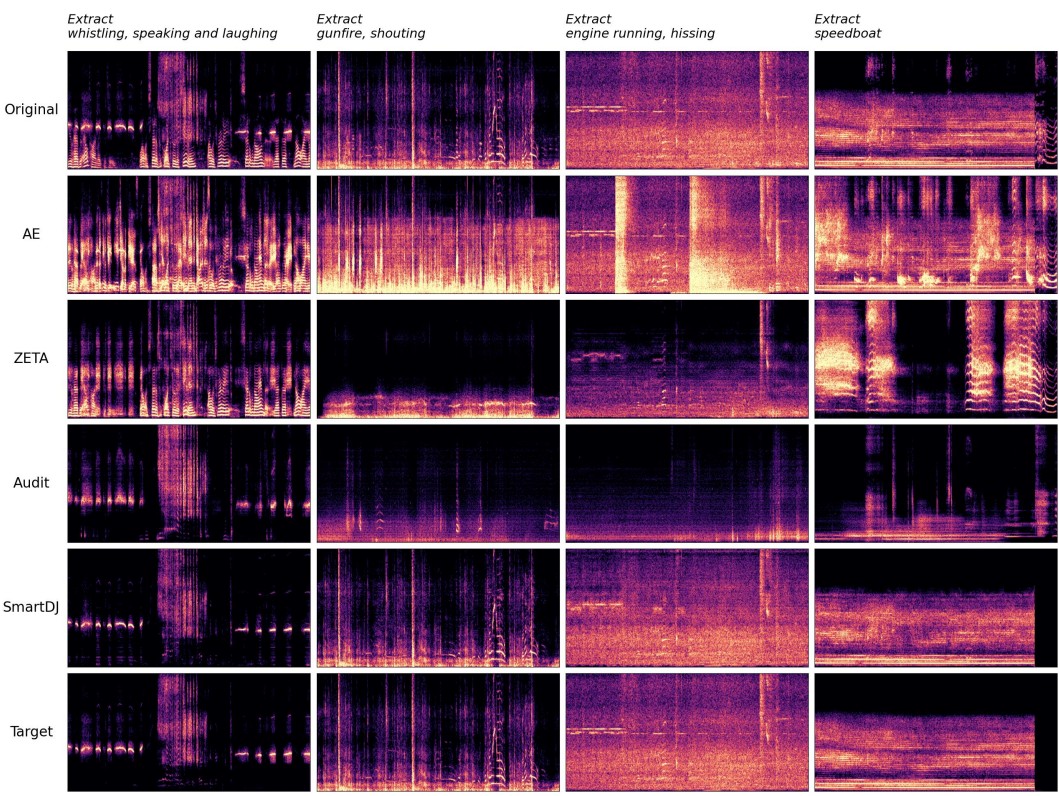

Figure 12: Examples on *extract* operation. The top row is the original audio and the bottom row is the target audio. Only SmartDJ can extract wanted audios that are clean and of high quality.

## C.4 HUMAN SUBJECTIVE STUDIES

We provide users with data pairs consisting of the original audio, the editing instruction, the SmartDJ edited result, and the edited results from a random competing baseline. In the case of single-step audio editing tasks *remove, extract, turn up/down, change sound direction* where there exists a ground-truth editing solution, we also provide it as the reference audio as shown in Fig. 13b. We conduct evaluations with 19 participants and 20 (10 complex audio editing, 10 single-step editing) data pairs per participant.

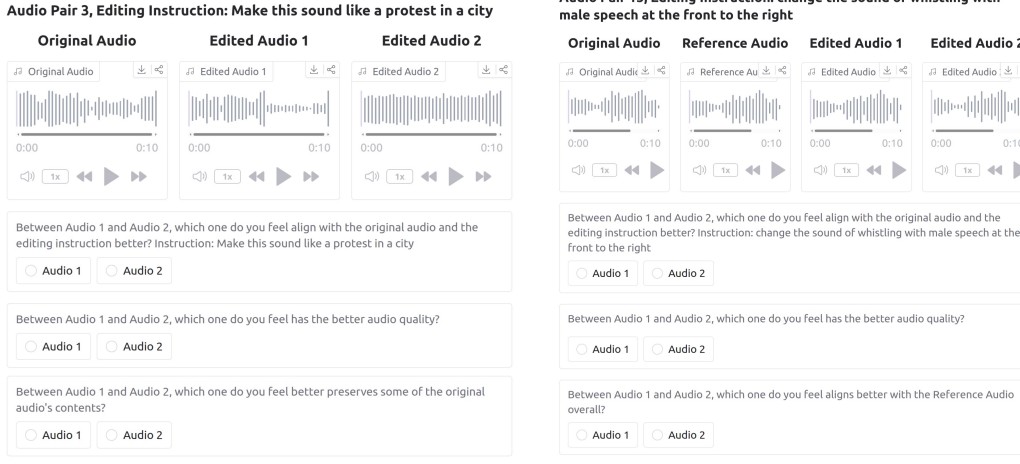

(a) User interface for complex audio editing task     (b) User interface for change direction editing task

Figure 13: The audio pairs, questions, and user interfaces for different audio editing tasks

In the 10 complex audio editing question pairs, we ask the user to select between the SmartDJ edited audio and the edited results from a random competing baseline (randomly sampled from AudioEditor, ZETA and Audit) according to the following three questions:

> **Question list for complex audio editing user study**
>
> - Between Audio 1 and Audio 2, which one do you feel aligns with the original audio and the editing instruction better?
> - Between Audio 1 and Audio 2, which one do you feel has the better audio quality?
> - Between Audio 1 and Audio 2, which one do you feel better preserves some of the original audio's contents?

In the single-step editing for *add* operation (3 pairs in total), we compare SmartDJ with a randomly sampled method from three baselines (AudioEditor, ZETA and Audit). We ask the user to answer four questions, with one additional question listed below:

> **Additional question for *add* operation user study**
>
> - Between Audio 1 and Audio 2, which one do you feel the added spatial effect aligns with the text instruction?

For the single-step *remove* and *extract* tasks (4 pairs) we compare SmartDJ with a randomly chosen baseline model. For *turn-up/turn-down* and *change direction* (3 pairs) we benchmark only against AUDIT, since the other baselines cannot perform these operations. Each pair includes a ground-truth reference, and listeners answer the three evaluation questions listed below.

> **Questions list for the *Remove, Extract, Turn up/down, Change sound direction* operation user study**
>
> - Between Audio 1 and Audio 2, which one do you feel aligns with the original audio and the editing instruction better?

> - Between Audio 1 and Audio 2, which one do you feel has the better audio quality?
> - Between Audio 1 and Audio 2, which one do you feel aligns better with the Reference Audio overall?

Across all tasks, SmartDJ is consistently preferred over all baseline methods by analyzing its win rate against three baselines. For the complex audio editing task, SmartDJ receives at least 80% of user votes over the baselines for audio quality, and at least 87% for alignment with the Declarative editing instruction and original audio. This shows that SmartDJ faithfully performs the requested scene transformation while preserving the key elements of the original audio. In the single-step editing task, our method receives more than 77% of user votes for audio quality, and 84% for alignment with the single-step editing text prompt, spatial description, and original or reference audio (when applicable). These results demonstrate that SmartDJ achieves the highest user preference across both quality and alignment metrics, outperforming all competing methods.

| Task & Metric | Baseline | Preference (SmartDJ) | 95% Confidence Interval | $p$-value |
|---|---|---|---|---|
| Declarative Edit Quality | ZETA | 80.00% | [68.9% − 91.1%] | < 0.001 |
| | AE | 95.50% | [90.6% − 100.0%] | < 0.001 |
| | Audit | 90.40% | [83.7% − 97.2%] | < 0.001 |
| Declarative Edit Alignment | ZETA | 87.00% | [80.4% − 93.6%] | < 0.001 |
| | AE | 91.00% | [86.2% − 95.9%] | < 0.001 |
| | Audit | 93.20% | [89.1% − 97.2%] | < 0.001 |
| Single Step Quality | ZETA | 77.40% | [66.1% − 88.6%] | < 0.001 |
| | AE | 86.30% | [76.8% − 95.7%] | < 0.001 |
| | Audit | 95.30% | [90.9% − 99.8%] | < 0.001 |
| Single Step Alignment | ZETA | 88.90% | [83.4% − 94.4%] | < 0.001 |
| | AE | 84.90% | [78.7% − 91.2%] | < 0.001 |
| | Audit | 96.80% | [94.2% − 99.3%] | < 0.001 |

Table 5: User Preference Evaluation Results.

Regarding the statistical significance of the user study results, we calculated it using a two-sided Z-test against a null hypothesis of random chance (50%). A p-value of < 0.001 indicates that there is less than a 0.1% probability that SmartDJ's preference rates are due to random luck. This confirms that the preference for our method is statistically significant. While the p-value confirms significance, the Confidence Interval demonstrates reliability. The 95% CI indicates the range in which the true population preference lies with 95% certainty. Because the lower bound of our confidence intervals is consistently well above 50% (ranging from 66.1% to 94.2%), we can claim with high certainty that SmartDJ is preferred over baselines in the general population.

## D USE OF LARGE LANGUAGE MODELS.

Large language models (LLMs) were used to support the writing of this manuscript. In particular, we used LLMs to assist with writing fluency, polishing the presentation of technical content, and rephrasing sections to improve readability. All scientific insights, technical innovations, and claims presented in this paper are solely the work of the authors. The final content was carefully reviewed, validated, and revised by the authors to ensure both correctness and clarity.

