# OpenReview forum: "SmartDJ: Declarative Audio Editing with Audio Language Model"
_ICLR.cc/2026/Conference — ICLR 2026 Poster_

### Official Review · Reviewer_NKMZ · 2025-10-15

**Soundness:** 3
**Presentation:** 2
**Contribution:** 3
**Rating:** 6
**Confidence:** 4

**Summary:**

This paper introduces SmartDJ, a novel framework for stereo audio editing that combines the reasoning capability of audio language models with the generative power of latent diffusion.

**Strengths:**

1. Novel Task Formulation: The paper introduces declarative editing for stereo audio, an area that remains relatively underexplored and represents a meaningful contribution to the field.

2. Valuable Dataset Contribution: The newly collected SmartDJ dataset provides a solid foundation for training and evaluating complex audio editing tasks, facilitating future research in declarative audio manipulation.

3. Effective Methodology: The proposed ALM-assisted pipeline demonstrates promising performance in handling complex editing scenarios.

**Weaknesses:**

1. I note that the paper primarily uses mono-channel audio clips for SmartDJ data synthesis. However, it lacks sufficient details on how spatial audio effects—such as binauralization or room acoustics (e.g., reverberation)—are implemented or considered.

2. Due to the cascaded architecture and separate training strategy, the performance of declarative editing heavily relies on the accuracy of the generated atomic instructions. It would be valuable to include quantitative results on the accuracy/feasibility of atomic instruction generation.

3. The connection between the proposed declarative audio editing and precise spatial effects manipulation remains unclear. For instance, in Example 1 (Appendix C.1), the instruction "Make this sound like a quiet afternoon in a garden" is fulfilled by adding a gentle breeze, but the spatial direction of this sound seems arbitrary.

4. The novelty of the single-step editing part appears limited, as its training protocol and model architecture closely follow those established in prior works. I am somehow confused.

5. The citation style does not conform to ICLR guidelines; mixing \citet{} and \citep{} reduces readability.

6. The only stated distinction between stereo and general audio editing is “direction,” yet only three direction types are considered. More directional categories are expected.

7. The dataset description is insufficient. “We sample 2–5 audio events and use GPT-4o to create 50k training pairs and 1k evaluation pairs of high-level audio editing data to train an audio language model and evaluate the editing pipeline.” This suggests the data are entirely generated and curated by GPT-4o. The evaluation set should be verified by human annotators for accuracy.

8. There is no comparison with existing editing datasets. Given the proposed 50k examples, please provide comparisons for both training and evaluation sets against prior editing datasets.

9. There are actually training-free editing methods, like DDPM inversion or better methods, that can be easily implemented for stereo editing. The authors should compare these models with SmartDJ.

**Questions:**

I will reconsider my score if my concerns are thoroughly addressed.

---

> ### Author Response · Authors · 2025-11-21
> **Author Response (1/2)**
>
> We would like to thank you for the valuable feedback and constructive suggestions. We are glad that you found our work has “novel task formulation” with “valuable dataset contribution” and “effective methodology”.  We address your concerns as follows:
>
> ---
> > **W1: Details on the Spatial Audio Effects.**
>
> The spatial cues are rendered using PyRoomAcoustics, where we simulate a stereo microphone pair with directional patterns. This introduces realistic ITD/ILD differences caused by the left and right ear. We also add the support of “add reverberation" operation (details in the later response) where we convolve the room impulse response generated also through PyRoomAcoustics. We will add these explanations in Sec. 3.5 and Appendix A.3 in the revised manuscript.
>
>
> ---
> > **W2: Include Quantitative Results on the ALM generated instruction.**
>
> We evaluate ALM’s ability to interpret declarative instructions and generate accurate atomic editing steps. We quantitatively evaluate the quality of the ALM-generated editing steps by comparing them with ground-truth editing steps. We include quantitative results to evaluate the bert score between the estimated instructions and the ground truth instructions. Our model achieves an average F1 score of 91.5%. This demonstrates that the ALM can reliably decompose declarative instructions into meaningful and accurate edit operations.
>
>
> ---
> > **W3: Connection between Proposed Declarative Audio Editing and Precise Spatial Effects .**
>
> Declarative editing in SmartDJ is driven primarily by semantic reasoning. The ALM determines what sound events should be added, removed, or adjusted to satisfy a high-level instruction. Declarative prompts (e.g., “a quiet afternoon in a garden”) do not explicitly specify spatial placement, so the ALM selects a reasonable but non-unique direction. This is expected, since pure text often underspecifies precise spatial intent.
> At the same time, spatial audio editing is a fundamental capability of our system. The audio editor supports explicit stereo manipulations, which gives users direct control over spatial location. Looking ahead, this spatial editing ability makes SmartDJ naturally extensible. Incorporating additional modalities including images or video that contain explicit visual spatial cues would allow us to fully ground event placement rather than relying solely on text. We will clarify this in the paper.
>
> ---
> > **W4: Discussion on the Novelty of Editing Models.**
>
> The main novelty of SmartDJ lies in the overall declarative editing framework. We are the first to propose combining an ALM planner with a stereo diffusion editor to execute multi-step editing from declarative instructions. This functionality is not supported by prior mono or template-based editors. Our semi-automatic dataset generation pipeline that produces declarative instructions at scale is also a core component enabling this framework.
> Regarding the diffusion editor, we use a standard DiT backbone and make it to support stereo operations. Due to this modular framework, we envision more advanced backbones/modules (e.g. multimodal conditions like image and video) for the audio editor could be plugged in to further enhance editing quality.
>
> ---
> > **Review W5: Citation Style.**
>
> We appreciate the reviewer for pointing this out. In the camera-ready version, we will standardize all citations to the ICLR-recommended style and remove the mixed use of \citet{} and \citep{} to improve readability.

---

> > ### Author Response · Authors · 2025-11-21
> > **Author Response (2/2)**
> >
> > ---
> > > **Review W6: Expanding more spatial direction on the atomic operations.**
> >
> > In the revised version, we expand the SmartDJ to support more than three spatial categories. The spatial direction is now rendered at multiple azimuths including left, left-front, front, front-right, right. In addition, we have expanded the editing vocabulary of SmartDJ to support more expressive manipulations. Specifically, our stereo-aware editor adds the following operations in addition to add, remove, extract and direction/volume adjustment in the original paper:
> > - Time shift: adjusting the sound timing by x seconds
> > - Add reverberation to a single source source:
> >     - Low reverberation level: convolve with RIR with RT60 of 0.4s
> >     - Mid reverberation level: convolve with RIR with RT60 of 0.8s
> >     - High reverberation level: convolve with RIR with RT60 of 1.2s
> > - Timbre adjust:
> >     - Bright: boost high frequency >3kHz
> >     - Dark: reduce high frequency >3kHz
> >     - Warm: boost components between 300Hz~4kHz and
> >     - Cold: boost components >6kHz and reduce components <300Hz
> >     - Muffled: reduce the component >1.5kHz
> >
> > These extensions enable SmartDJ to handle more diverse and broader editing scenarios beyond our previous operations. We also evaluate these new operations in the following table. SmartDJ consistently improves over the Audit baseline. Additionally, we provide new examples of these operations in the updated project page of supplementary materials.
> >
> >
> > | Operation  | Method|FD|FAD|KL|  LSD |  GCC  |  CRW  | FSAD |
> > |-|-|:-:|:--:|:-:|:-:|:--:|:-:|:-:|
> > | Shift time |  Audit  | 31.91 | 5.11 | 1.35 | 1.31 | 127.9 | 125.9 | 0.87 |
> > || SmartDJ | 10.01 | 1.25 | 0.40 | 1.12 |  1.82 | 25.96 | 0.10 |
> > |Timbre| Audit  | 31.71 | 3.91 | 1.40 | 1.09 | 135.2 | 135.9 | 0.84 |
> > ||SmartDJ |  8.68 | 0.80 | 0.35 | 1.04 |  3.87 |  32.2 | 0.11 |
> > |Rever |  Audit  | 31.42 | 3.68 | 1.45 | 1.06 | 132.1 | 127.5 | 0.81 |
> > |  | SmartDJ | 10.20 | 0.98 | 0.47 | 1.01 |  6.67 | 23.16 | 0.12 |
> >
> >
> >
> >
> > ---
> > > **W7: Include Human Validation of Evaluation Set.**
> >
> > Thank you for raising this important question. Our dataset generation is semi-automatic, not fully GPT-curated. GPT-4o is used only to propose declarative instructions and draft atomic steps; all audio is synthesized through a deterministic signal-processing pipeline. For the evaluation set, we perform additional human verification stages. We manually filter out cases where GPT-4o hallucinated nonexistent events or produced inconsistent edit steps, and we remove noisy or low-quality audio mixtures that would make CLAP score evaluation unreliable. The final 1k evaluation set therefore contains only valid, and semantically consistent editing trajectories. We will clarify this process in Sec. 3.5 and Appendix A.
> >
> >
> > ---
> > > **W8: Comparison of dataset with prior editing datasets.**
> >
> > Following prior work Audit, the training set of 50K examples is from a mixed corpus of AudioCaps, VGGsound, FSD50k, ESC50 and WavCaps. And the rest of the testing set of 1K examples are from the test set of AudioCaps and VGGSound. We will make it clear in the revised version.
> >
> > ---
> > > **W9: include evaluations of training-free methods for spatial location change.**
> >
> > We appreciate the reviewer’s suggestion to compare against training-free editing methods. We would like to clarify that our evaluation baselines include the audio adaptations of these specific training-free methods. We implement them on two stereo audio backbones: Stable Audio Open 2.0 and Both Ear Wide Open (BEWO).
> >
> > - Standard Inversion (DDIM/DDPM): We included baseline inversion methods like ZETA (based on DDPM Inversion), SDEdit, and standard DDIM Inversion that primarily focus on editing the input signal with global noise-mixing.
> >
> > - Prompt-to-Prompt (P2P): Our AudioEditor baseline represents the Prompt-to-Prompt (P2P) paradigm. The method implements P2P's core contribution, Cross-Attention Control, and a further Null-Text Inversion.
> >
> > In addition, we include experiments on stereo editing metrics (changing sound direction). SmartDJ outperforms all training-free editing methods. We will include the above explanation and analysis in our revised paper to avoid misunderstandings.
> >
> > | Method  |   FD  |  FAD |  KL  |  LSD |   GCC  |   CRW  | FSAD | FSAD |
> > |---------|:-----:|:----:|:----:|:----:|:------:|:------:|:----:|:----:|
> > |  SDEdit | 41.47 | 3.83 | 1.78 | 1.42 |  165.6 | 169.56 | 0.49 | 0.49 |
> > |   DDIM  |  58.0 | 7.32 | 2.85 | 1.30 | 193.71 | 163.92 | 0.17 | 0.17 |
> > |   ZETA  |  39.6 | 3.35 | 1.38 | 1.54 | 158.78 | 173.81 | 0.42 | 0.42 |
> > |    AE   |  46.5 | 4.95 | 1.82 | 1.51 |  95.04 | 133.12 | 0.32 | 0.32 |
> > | SmartDJ |  13.0 | 0.88 | 0.33 | 1.00 |  59.6  |  36.6  | 0.02 | 0.02 |

---

> > > ### Comment · Reviewer_NKMZ · 2025-11-21
> > > **Reply to Submission422 Authors**
> > >
> > > At first, I really liked this paper and thought it had a reasonable motivation and contribution. However, I believe it still lacks important components and sidesteps many critical issues.
> > >
> > > > W1: Details on the Spatial Audio Effects.
> > >
> > > The method of implementing spatial audio is extremely important. It should address multiple questions, such as interaural distance and microphone orientation. These factors are crucial for the realism of the simulation. However, I still cannot find any information about this in the revised paper. I’m not sure whether the authors are unaware that ICLR requires revisions at this stage. In any case, I still know nothing about it.
> > >
> > > > W2: Include Quantitative Results on the ALM generated instruction.
> > >
> > > How are quantitative results used to evaluate the “bert” score between the estimated instructions and the ground-truth instructions? Regardless of whether “bert” might actually be the mistaken “best,” I still don’t know how it was evaluated. Was a model used, or was it done manually? If it were manual, what was the evaluation strategy?
> > >
> > > > W5: Citation Style.
> > >
> > > You actually can revise the paper right now.
> > >
> > > > W6: Expanding more spatial direction on the atomic operations.
> > >
> > > I don't know how you extend to more than three spatial categories. No description, experiment, example, and evaluation.
> > >
> > > > W8: Comparison of the dataset with prior editing datasets.
> > >
> > > My question concerns the data quantity and data format compared with other editing datasets, which is crucial for a paper that includes a dataset. However, the authors provided the distribution of the source dataset instead. These are two completely different concepts. I don't think 50K is enough for this field compared to editing datasets or simulated datasets. Thanks to the automated baseline, I think you are able to make it 500K or 5M.
> > >
> > > I like this topic, but somehow I feel disappointed. Overall, I think this is an 8-point topic in a 4-point paper. I decrease my score to 4 until it gets better.

---

> > > > ### Author Response · Authors · 2025-11-23
> > > > **Further Author Response (1/2)**
> > > >
> > > > Thank you very much for your prompt response and detailed feedback. We have carefully addressed each of your concerns and revised both the main manuscript and the appendix accordingly. All the changes are highlighted in the color blue. Below, we provide point-by-point clarifications and indicate where the corresponding changes appear in the paper. We genuinely appreciate your rigorous review and constructive input, which have substantially strengthened the quality and clarity of our work.
> > > >
> > > > ---
> > > > > **Follow up on W1: “The method of implementing spatial audio is extremely important.”**
> > > >
> > > > Thank you for emphasizing the importance of implementation details for spatial audio. We have updated the manuscript and included more implementation details about acoustic room impulse response simulation in the Sec 4.1 setup (with subtitles “Datasets” and “Spatial Sound Effects”) and Appendix Sec A.2 (with the subtitle “Detailed Implementation of Spatial Audio”), which we summarize below.
> > > >
> > > > Our binaural RIR simulation uses a distance of 0.17m between users’ ears to simulate the interaural time difference and adopts a cardioid directivity gain pattern to model left/right ear differences when sound events are coming from different directions. To introduce room realism, we further simulate reverberation using RT60 ranges from 0.3s to 1.2s. Room size, microphone orientation, and source/microphone placement are randomized during synthesis. To render the left and right ear received signal, we synthesize their corresponding RIR h(t) and convolve with the dry sound to get the final wet sound that creates spatial audio effect. This h(t) considers the gain pattern effect by applying different weights when combining acoustic rays from different directions.
> > > >
> > > >
> > > > ---
> > > > > **Follow up on W2: “How are quantitative results used to evaluate the “bert” score between the estimated instructions and the ground-truth instructions? Regardless of whether “bert” might actually be the mistaken “best,” I still don’t know how it was evaluated. Was a model used, or was it done manually? If it were manual, what was the evaluation strategy?”**
> > > >
> > > > Thank you for pointing out the need for clearer explanation on the “BERT” score. We have clarified this ALM evaluation procedure in the revised paper in the ablation study part (Sec. 4.3 with the subtitle “Performance of ALM generate editing steps”). We summarize our revised content as follows.
> > > >
> > > > We use BERTScore [1], a metric for natural language generation that measures semantic similarity between two text sequences using contextual embeddings from a pretrained BERT model. We follow the official implementation: https://github.com/Tiiiger/bert_score.
> > > >
> > > > For our task, each detailed editing plan consists of multiple atomic steps. To ensure fair evaluation, we group atomic operations by type (add, remove and so on), and compute BERTScore only between matching operation types in the ground-truth plan and the ALM-generated plan. For example, ground-truth “add” steps are concatenated together compared only against generated and concatenated “add” steps, and similarly for “remove” and other operations. If the generated or ground truth operation does not contain a specific operation, it will be replaced by a “[MASK]”. We then compute BERTScore for each operation type and report the average across all operation groups (results in 91.8% Precision, 92.0% Recall and 91.5% of F1). This provides a model-based quantitative measure of how closely the ALM’s generated steps match the semantics of the ground-truth atomic instructions.
> > > >
> > > > [1] Zhang et al. 2020. "BERTScore: Evaluating Text Generation with BERT." ICLR 2020.
> > > >
> > > >
> > > > ---
> > > > > **Follow up on W5: “Citation style”**
> > > >
> > > > We have revised the paper with a consistent reference style.

---

> ### Author Response · Authors · 2025-11-23
> **Further Author Response (2/2)**
>
> ---
> > **Follow up on W6: “More explanation on the spatial direction”**
>
> Thank you for your comments. We have revised the manuscript and included more details in Sec 4.1 (with subtitles “Datasets” and “Spatial Sound Effects”) and Appendix Sec A.2 (with the subtitle “Detailed Implementation of Spatial Audio”). We summarize as follows.
>
> We have expanded the spatial categories beyond the original three. SmartDJ now supports more discrete stereo directions: Left (180°), Right (0°), Center (90°), Front-Left (135°), Front-Right (45°), with random perturbation (Gaussian noise with standard deviation of 5°). To support these new spatial directions and new editing operations (timbre, time, reverb), we re-synthesized additional single-step editing samples (now 1M in total) with these expanded spatial labels and fine-tuned the diffusion editor accordingly. New demos are included in the project page of newly updated supplementary material. Our new experiment results have just been updated in the revised manuscript in Table 2.
>
> ---
> > **Follow up on W8: “My question concerns the data quantity and data format compared with other editing datasets, which is crucial for a paper that includes a dataset. However, the authors provided the distribution of the source dataset instead. These are two completely different concepts. I don't think 50K is enough for this field compared to editing datasets or simulated datasets. Thanks to the automated baseline, I think you are able to make it 500K or 5M.”**
>
> Thank you for raising this point. We want to clarify that we use two datasets during training:
>
> - (1) Single-step audio editing dataset.
> This dataset is used to train the diffusion-based audio editor from scratch. Building this dataset is easy through audio signal processing. In the latest revision, we have expanded this dataset to include 1M training samples supporting additional operations and expanded spatial directions.
>
> - (2) Declarative audio editing dataset (for ALM instruction finetuning).
> This dataset is used to fine-tune the audio language model to reason about the high-level command together with audio and generate atomic editing steps. This audio language model is pretrained using a large corpus of audio question-answering dataset (4.2M audio-question-answering pairs) [2,3]. This pretrained model shows strong audio understanding and reasoning capabilities. We believe that with our high-quality dataset, it requires much less data to finetune this ALM than training the diffusion-based editor. It is also common to finetune LLM models with relatively small datasets [4,5].
>
> Finally, we will fully open-source our dataset synthesis pipeline, including both the single-step audio editing and declarative audio editing, to the community so that researchers can easily reproduce our results and explore larger or customized datasets.
>
> [2] Kong et al. "Audio Flamingo: A Novel Audio Language Model with Few-Shot Learning and Dialogue Abilities." ICML 2024
>
> [3] Ghosh et al. "Audio Flamingo 2: An Audio-Language Model with Long-Audio Understanding and Expert Reasoning Abilities." ICML 2025
>
> [4] Yang et al. "Smalltolarge (s2l): Scalable data selection for fine-tuning large language models by summarizing training trajectories of small models." NeurIPS 2024
>
> [5] Wang et al. "Data whisperer: Efficient data selection for task-specific LLM fine-tuning via few-shot in-context learning." ACL 2025.

---

> ### Comment · Reviewer_NKMZ · 2025-11-23
> **Reply to Submission422 Authors**
>
> Although I carefully reread the author’s rebuttal, there are still some issues that cannot be overlooked.
>
> The authors claim that the dataset size increased from 50k in the submitted version to 1M in the rebuttal version within just two days. I’m not sure whether this 20× increase introduced too much new content, or how the quality of the 1M samples was ensured within two days.
>
> All I want to see is a quantitative comparison and distribution comparison of the proposed dataset, similar to Table 2 in Data Whisperer. I’m quite sure you’ve read it, since you cited it in your last rebuttal. That kind of table is the most common format for comparisons. However, I still cannot find any dataset comparison table in the paper. You need to summarize all the related datasets of the related task for comparison in a single table. This table highlights the differences in tasks as well as in data quantity and data format.
>
> I appreciate the authors’ hard work. However, I have already spent sufficient time going through all the rebuttals. Since there are remaining concerns, my final score will depend on the discussion among the reviewers.

---

> > ### Author Response · Authors · 2025-11-25
> > **Further Author Response**
> >
> > Thank you for your prompt follow-up comments. We are happy that our previous rebuttals can help solve your concerns on the spatial audio implementation and ALM evaluation. Below, we would like to further clarify our dataset compositions and the distinction between the two types of dataset in our paper below:
> >
> > - (1) Single-step audio editing dataset (1M samples).
> > This dataset is used to train the diffusion-based audio editor only. It is created entirely through audio processing and simulation, without GPT involvement. To avoid potential misunderstandings, we clarify that at the time of the paper’s initial submission, the dataset size was 500k, as stated in the paper Sec. 4.1, Line 310. Due to the fast signal processing pipeline, we were able to scale it from 500K to 1M for the past week and further fine-tune our diffusion audio editor to support more spatial directions and more audio editing operations. We subsequently updated Table 2 in the paper on the single step audio editing.
> >
> > - (2) Declarative audio editing dataset (50k samples).
> > This dataset is used exclusively for ALM instruction fine tuning. Each data sample contains: a declarative editing instruction, the original audio, the decomposed single-step editing plans and the edited audios. This dataset is smaller than the single step audio editing dataset because its purpose is to fine-tune the audio language model. As we mentioned, the ALM starts from a very strong pretrained backbone and it requires less data to finetune it. Nevertheless, we are currently expanding this dataset to have 250k samples and training the ALM planner to support more diverse editing operations. We will have more results/demos in the coming days before the end of rebuttal.
> >
> > ---
> > > Dataset comparison table.
> >
> > Thank you for bringing up the dataset comparison table and this would help us gain a more clear view about the dataset. We have added a dataset comparison in the revised appendix (table 4) summarizing relevant existing audio editing and generation datasets.

---

> ### Comment · Reviewer_NKMZ · 2025-11-25
> **Reply to Submission422 Authors**
>
> Thank you for your response. That answers my question.
>
> While it would be preferable to include Table 4 in the main paper, I have adjusted my score accordingly.
>
> I look forward to using your code and data.
>
> Best of luck!

---

> > ### Author Response · Authors · 2025-11-26
> >
> > Thank you very much for all your comments and review and for adjusting your score. We sincerely appreciate your careful evaluation and your positive feedback. We are glad that our clarification addressed your question, and we look forward to sharing the code and data after camera ready.
> >
> > Thank you again for your time and for helping strengthen our submission.

---

> > > ### Author Response · Authors · 2025-12-03
> > > **Follow-up on expanding declarative audio editing dataset**
> > >
> > > We have scaled up the dataset scale to 250K and have some preliminary results/demo on the dataset. We will include additional full experiments and demos in the camera-ready version. Nevertheless, the current results on the new 250K dataset still shows similar results. When comparin full Audit + SmartDJ ALM pipeline against SmartDJ alone, our method remains strong across all metrics. We also updated some new declarative audio editing examples in the supplementary material of demo page.
> > >
> > > |Variant|FD|FAD|KL|LSD|IS|CLAP|
> > > |:-:|:-:|:-:|:-:|:-:|:-:|:-:|
> > > |Audit + SmartDJ ALM|16.5|4.41|1.67|1.40|5.63|0.213|
> > > |SmartDJ|12.7|1.33|1.43|1.35|9.36|0.241|

---

### Official Review · Reviewer_QgiL · 2025-10-16

**Soundness:** 3
**Presentation:** 3
**Contribution:** 3
**Rating:** 4
**Confidence:** 5

**Summary:**

This paper presents a framework for high-level audio editing that allows users to modify stereo audio through semantic instructions rather than low-level waveform manipulation. The system introduces an Audio Language Model (ALM) that generates step-by-step editing commands, guiding an audio diffusion model to perform multi-stage modifications until an optimal output is reached. The paper explores the feasibility of such semantic audio editing and evaluates its ability to handle complex instructions such as spatial or environmental sound changes.

**Strengths:**

•  The paper addresses a novel and forward-looking task — semantic, high-level audio editing — which goes beyond traditional waveform-level operations.

•  The idea of integrating an Audio Language Model that provides editing steps is creative and can inspire future work combining reasoning and audio generation.

•  The model supports stereo audio editing, which is technically more challenging and rarely explored in prior works.

•  The volume control (±dB) feature is interesting, as most baseline editing systems cannot precisely handle such parametric changes.

•  The overall direction contributes to bridging the gap between natural-language interaction and multi-step sound manipulation, a promising area for interactive audio systems.

**Weaknesses:**

•  Although the task is conceptually interesting, the training and testing data are artificially concatenated, which raises concerns about the naturalness and generalizability of the results. Real-world editing tasks often involve continuous, unstructured recordings rather than segmented mixtures.

•  The proposed framework mainly focuses on environmental or ambience-level edits (e.g., changing the spatial context or reverb), while many practical audio-editing scenarios involve precise, localized edits (e.g., removing or amplifying a specific sound event), which existing baseline models can already handle effectively.

•  The system architecture resembles a captioning + LLM planning + diffusion editing pipeline. The novelty is somewhat limited since similar modular approaches could be implemented by combining existing captioning and LLM-based reasoning models.

•  The Audio Language Model (ALM) component’s added value is unclear — it might be replaceable by prompting a strong general LLM (e.g., ChatGPT, Llama 3) with audio captions and editing prompts to produce similar step-wise operations.

•  The diffusion-based editor uses a standard DiT backbone without architectural innovation; exploring more modern generative frameworks like flow-matching or bridge-flow models could potentially improve performance and efficiency.

•  Some tables and metrics (e.g., Table 1) are difficult to interpret — it is not clear whether they measure end-to-end editing quality or only the performance of the editor module， especially it seems that there are a lot of new editing instructions generate by the ALM, such as in/decreasing the volume by dB.

**Questions:**

•  How does the proposed LDM-based editor compare to existing audio separation or editing models in quantitative and perceptual metrics? Only on the editing level, not combining with the ALM guided steps.

•  What is the dataset used on result table 2, can we compare the results on the same evaluation set used by Audit?

•  Can the ALM component be replaced by an audio-captioning model + LLM prompting pipeline? If not, what specific advantages does ALM bring?

•  Have the authors considered evaluating intermediate steps of ALM-guided editing (e.g., SmartDJ examples) to show how the step-wise process improves the result? Currently the demo only show the comparision on the final results.

---

> ### Author Response · Authors · 2025-11-21
> **Author Response (1/2)**
>
> We would like to thank you for the thoughtful review and constructive feedback. We are glad that you found our work “addresses a novel and forward-looking task” and “inspires future work”.  We address your concerns as follows:
>
> ---
> > **Concerns about Dataset and Adaptation to Real-world Continuous Recording: “Although the task is conceptually interesting …rather than segmented mixtures.”**
>
> Regarding the concern on "continuous, unstructured recordings”, the audio editing task assumes that the user already provides a recorded audio clip. This is common in professional workflows in sound design and post-production. In this scenario, editing typically starts from a fixed audio recording rather than an unbounded audio stream. SmartDJ is designed for this setting.
>
> Regarding “training/evaluation data are artificially concatenated”, although the dataset is synthesized, it is not composed of ideal clips without noise. Besides, when multiple raw clips are mixed, the resulting audio could contain overlapping audio events. This produces cluttered mixtures that mirror natural ambient environments.
>
> To further assess real-world generalization, we also provide demos with audio clips from YouTube in the project page of supplementary material. SmartDJ works successfully on these samples. This demonstrates that the system can generalize to real world audio recordings.
>
> ---
> > **Focus of the Framework on Ambience-Level Edits and Localized Edits: “The proposed framework mainly focuses on environmental … can already handle effectively.”**
>
> We would like to clarify that SmartDJ is not limited to ambience-level adjustments. The main motivation of our work is to enable declarative editing, which traditional template-based editors would fail. However, the audio editor in our framework is explicitly designed for localized event-level edits, such as add, remove, extract, turn up/down, and change direction, each of which targets individual sound events. In addition, we expanded the editing vocabulary editor to support more manipulations including:
> - Time shift: adjusting the sound timing by x seconds
> - Add reverberation to a single source source:
>     - Low reverberation level: convolve with RIR with RT60 of 0.4s
>     - Mid reverberation level: convolve with RIR with RT60 of 0.8s
>     - High reverberation level: convolve with RIR with RT60 of 1.2s
> - Timbre adjust:
>     - Bright: boost high frequency >3kHz
>     - Dark: reduce high frequency >3kHz
>     - Warm: boost components between 300Hz~4kHz and
>     - Cold: boost components >6kHz and reduce components <300Hz
>     - Muffled: reduce the component >1.5kHz
>
> We also provide new examples of these operations in the updated project page in the supplementary materials.
>
> |Operation|Method|FD|FAD|KL|LSD|GCC|CRW|FSAD|
> |:-:|:-:|:-:|:-:|:-:|:-:|:-:|:-:|:-:|
> |Shift time|Audit|31.91|5.11|1.35|1.31|127.9|125.9|0.87|
> | | SmartDJ |10.01|1.25|0.40|1.12|1.82|25.96|0.10|
> |Timbre|Audit|31.71|3.91|1.40|1.09|135.2|135.9|0.84|
> | | SmartDJ|8.68|0.80|0.35|1.04|3.87|32.2|0.11|
> |Reverb|Audit|31.42|3.68|1.45|1.06|132.1|127.5|0.81|
> | |SmartDJ|10.20|0.98|0.47|1.01| 6.67|23.16|0.12|
>
> As reported in Table 2 and this new table, SmartDJ significantly outperforms baselines on these localized editing. These results indicate that the system is fully capable of both (i) handling detailed, event-level edits and (ii) executing complex transformations guided by declarative instructions.
>
> ---
> > **What is the benefit of ALM, can it be replaced by an audio-caption model + LLM? : “The system resembles a captioning … LLM-based reasoning models.”**
>
> Our ALM module cannot be replaced by a captioning model combined with a generic LLM for the following reasons.
> - Captioning models lose event-level grounding. Audio captions typically summarize only the dominant audio events and do not preserve fine-grained events especially when overlapping happens. However, declarative editing requires precise event-level structure. The ALM learns this structure directly from audio and therefore retains far richer grounding than text captions.
> - Caption + LLM introduces cumulative error. Any captioning error, like missing an event, misidentifying a source, can propagate into the LLM's reasoning and compound across steps. This results in hallucinated or non-executable editing instructions (e.g., removing events that do not exist).
> Empirically, we implemented an alternative pipeline (audio captioning → LLM planning → diffusion editor), using the captioning model from DCASE audio caption challenge [1] to describe the audio and GPT-4o to generate step-wise instructions, which are executed by SmartDJ editor. Our quantitative comparison shows that the captioning-based pipeline performs consistently worse than SmartDJ across all metrics.
> |Variant|FD|FAD|KL|LSD|IS|CLAP|
> |:-:|:-:|:-:|:-:|:-:|:-:|:-:|
> |AudioCaption + GPT4o + SmartDJ editor|16.8|2.70|2.96|1.45|6.35|0.184|
> |SmartDJ|14.7|1.53|2.85|1.42|8.36|0.238|
>
> [1] https://github.com/wsntxxn/AudioCaption

---

> > ### Author Response · Authors · 2025-11-21
> > **Author Response (2/2)**
> >
> > ---
> > > **Table 1 is hard to interpret: “Some tables and metrics (e.g., Table 1) are difficult to interpret … such as in/decreasing the volume by dB.”**
> >
> > Table 1 reports end-to-end editing performance, measuring the final audio produced by the full declarative pipeline. In this pipeline, ALM generates a sequence of atomic operations and the diffusion editor executes them sequentially. All metrics in Table 1 therefore reflect the quality of the complete edited output, not the performance of individual steps. In contrast, Table 2 evaluates the single-step editor in isolation: each atomic operation (e.g., add, remove/extract, volume change, change direction) is applied independently in a controlled setting to disentangle the editor’s capability from the ALM’s reasoning accuracy. We will revise the manuscript to make this separation more explicit and ensure that readers can easily interpret both evaluation levels.
> >
> > ---
> > > **Comparison between LDM-editor and existing audio separation model: “How does the proposed LDM-based editor compare to … ALM guided steps.”**
> >
> >
> > We directly compare our LDM-based editor against an existing powerful separation model. In the ablation study, we evaluate SmartDJ on the Extract (similar to audio separation) operation against AudioSep [2]. As shown in the last row of Table 3 of our original manuscript, SmartDJ achieves comparable or slightly better performance across FD, FAD, and KL metrics. This demonstrates that even though our editor is designed for general-purpose stereo editing rather than separation alone, it remains competitive with dedicated separation models on targeted editing operations.
> >
> > We have also compared against the existing audio editing method Audit on single-step audio edits (without ALM guided steps). Our LDM-editor outforms Audit across all quantitative and perceptual metrics on all single-step editing types.
> >
> > [2] Liu, Xubo, Qiuqiang Kong, Yan Zhao, Haohe Liu, Yi Yuan, Yuzhuo Liu, Rui Xia, Yuxuan Wang, Mark D. Plumbley, and Wenwu Wang. "Separate anything you describe." IEEE/ACM Transactions on Audio, Speech, and Language Processing (2024).
> >
> > ---
> > > **Provide more details about the dataset on Table 2: “What is the dataset used on result … used by Audit?”**
> >
> >
> > Table 2 evaluates the single-step editor only. We synthesize 1k audio samples from the test set of AudioCaps with single-step audio editing operations. All the baseline models are evaluated on exactly the same single-step evaluation set. However, we can not directly compare the dataset with the evaluation set used by Audit, as they did not release more details on the testing dataset.
> >
> > ---
> > > **Provide demos on the intermediate steps of ALM-guided edits: “Have the authors considered evaluating intermediate steps … comparisons on the final results.”**
> >
> >
> > In the revised project page of supplementary material, we now include intermediate audio outputs from the ALM-guided editing process. Quantitatively evaluating each intermediate state is difficult because performance depends on both the atomic operation and its ordering, which varies across instructions. For fairness, we therefore evaluate each atomic operation in isolation. As shown in Table 2, SmartDJ achieves consistently stronger performance than all baselines on every single-step edit, indicating that each component operation is reliable and contributes to the improved end-to-end results shown in Table 1.

---

> > > ### Comment · Reviewer_QgiL · 2025-11-22
> > >
> > > Thank you very much for your response and for addressing many of the questions raised. However, several significant issues remain. For example, the fact that much of the data is manually stitched together is a serious concern. Although the model performs well on some real-world YouTube videos, these cases do not necessarily represent the general behavior of the method. In addition, the explanation regarding the captioning + LLM pipeline is still unconvincing. I continue to believe that a strong captioning model combined with an LLM could potentially replace much of the proposed framework.
> > >
> > > Overall, this is an interesting paper with promising results, but in my opinion, the level of novelty is not sufficient for acceptance at ICLR. Good luck with your submission.

---

> > > > ### Author Response · Authors · 2025-11-24
> > > > **Further Author Response**
> > > >
> > > > Thank you again for the thoughtful follow-up. We would like to briefly clarify the remaining concerns:
> > > >
> > > > ---
> > > > >**Clarification on "stitching" or "concatenating" audio "segments":**
> > > >
> > > > We would like to clarify that our dataset does *not* stitch or concatenate audio clips sequentially. Instead, when combining two or more sources, we mix them by adding their waveforms in the time domain, consistent with how real-world sounds physically combine. This linear superposition naturally produces overlapping acoustic events rather than “event A followed by event B,” and is the standard formulation used in audio signal processing. When adding multiple sound sources, we also apply random timing offset (between 0 to 3 seconds) to have natural temporal variation. In addition, we augment these real-world recordings with spatial room acoustic simulations (reverberation, source directivity, microphone placement, etc.), producing mixtures that closely resemble everyday audio environments. We have also updated the detailed explanation of room impulse response synthesis in Section 4.1 and Appendix A2.
> > > >
> > > > ---
> > > > >**On “audio captioning” + LLM as an alternative:**
> > > >
> > > >
> > > > Thank you for raising this point. We agree that if one has access to a sufficiently strong audio captioning model that can reliably describe dense, multi-source mixtures, then combining it with an LLM can serve as a reasonable alternative to our ALM-based decomposition. Motivated by this, we experimented with using AF2 (Audio-Flamingo 2) itself as a captioning model: we prompted AF2 to directly list the detailed audio events in the mixture and then used GPT-4o to convert those events into a sequence of editing actions. This captioning + LLM pipeline produces reasonable decompositions.
> > > >
> > > > |Variant|FD|FAD|KL|LSD|IS|CLAP|
> > > > |:-:|:-:|:-:|:-:|:-:|:-:|:-:|
> > > > |AudioCaption + GPT4o + SmartDJ editor|16.8|2.70|2.96|1.45|6.35|0.184|
> > > > |AF2 Caption + GPT4o + SmartDJ editor|15.9|1.73|2.94|1.44|7.39|0.208|
> > > > |SmartDJ|14.7|1.53|2.85|1.42|8.36|0.238|
> > > >
> > > > Nevertheless, AF2 is already an audio-language model trained on large audio corpora. When we fully fine-tune AF2 on our declarative editing dataset as an editing planner, we reduce error accumulation between captioning and decomposition steps. As shown in the table, the fully fine-tuned model consistently outperforms both the standard audio caption model and the AF2 captioner pipeline across all metrics. This demonstrates that our ALM-based approach is not simply a captioning+LLM reformulation, but a more reliable and editing-oriented solution.
> > > >
> > > >
> > > > ---
> > > > >**Novelty of the Paper:**
> > > >
> > > >
> > > > Thank you for raising the concern about novelty. We summarize the key contributions of our work below.
> > > >
> > > > 1. A new problem formulation: declarative audio editing. (by Reviewer NKMZ)
> > > > Prior audio editing systems require explicit step-by-step instructions. SmartDJ is the first system designed for declarative editing, where a single natural-language instruction must be parsed into both a scene decomposition and an ordered sequence of edits.
> > > >
> > > > 2. A unified ALM + diffusion framework for editing-oriented decomposition and execution. (by reviewer bDhl, eq8D, NKMZ)
> > > > Our method introduces a two-stage architecture designed specifically for declarative editing with a fine-tuned ALM that generates multi-step editing plans aligned with audio transformations. We incorporate a stereo diffusion editor trained to execute these plans with event-level control. This joint design enables declarative editing and human-in-the loop editing interactions.
> > > >
> > > > 3. A scalable data-generation pipeline enabling supervised training. (by reviewer NKMZ and bDhL)
> > > > Since declarative audio editing datasets did not exist, we developed a semi-automatic pipeline that generates tuples of (original audio, declarative instruction, decomposition, edited audio) at scale, making it possible to train both components in a supervised and consistent manner.
> > > >
> > > > Together, these contributions introduce a new problem setting, a unified framework for declarative audio editing, and the dataset infrastructure required to support it.

---

### Official Review · Reviewer_eq8D · 2025-10-23

**Soundness:** 3
**Presentation:** 3
**Contribution:** 3
**Rating:** 6
**Confidence:** 5

**Summary:**

This paper introduces a framework for audio editing that breaks down high-level (declarative) instructions into a series of basic editing steps, using an LDM model trained to carry out these edits. A data synthesis pipeline is employed to create paired datasets for training. Experimental results indicate this approach delivers better perceptual quality and greater alignment with user instructions compared to several baseline methods.

**Strengths:**

Innovative use of both Audio Language Models (ALM) and Large Language Models (LLM):
1. The ALM acts as a planner by understanding input audio, interpreting broad user instructions, and segmenting them into a sequence of simpler editing operations.
2. The LLM contributes as a designer during data creation, generating a wide range of high-level edit commands and their corresponding atomic steps.

The study includes comprehensive experiments and ablation analyses.

Allow editing of stereo audio direction.

**Weaknesses:**

Breaking down user instructions into atomic editing actions isn't a novel concept for audio editing; for instance, WavCraft already transforms commands into several subtasks.

Complicated instructions may not always be easily reducible to a fixed set of basic operations.

**Questions:**

Are there any additional atomic operations to consider beyond the current ones?

When using Audit as a baseline, do authors use the same training data? If so, what explains the significant subjective preference when comparing SmartDJ and Audit in single-step test?

---

> ### Author Response · Authors · 2025-11-21
> **Author Response (1/2)**
>
> Dear Area Chair and Reviewers,
>
> We sincerely thank the Area Chair for overseeing the review process and the reviewers for thoughtful assessments and overall positive evaluations. Below we first summarize the key comments from the reviewers, followed by the major revisions we have made to address the concerns.
>
> Our paper’s strengths were consistently highlighted across three main dimensions: novelty of the task, originality of the methodology, and strong empirical capabilities and results.
>
> **1. Novelty of the Task**
>
> Reviewer QgiL noted that our work “addresses a novel and forward-looking task,” emphasizing that declarative and stereo-audio editing is “technically more challenging and rarely explored in prior works.” Reviewer NKMZ similarly remarked on our “novel task formulation” and “valuable dataset contribution.”
>
>
> **2. Originality of our Method**
>
> Reviewer QgiL highlighted the “creative” nature of our approach and its potential to “inspire future work combining reasoning and audio generation.” Reviewer bDhL further praised the “strong originality” and “clever modular design,” a sentiment echoed by reviewer eq8D, who commended our “innovative use of ALM and LLM.”
>
>
> **3. Strong capabilities and results**
>
> Reviewer bDhL described our experiments as “comprehensive,” while reviewer eq8D appreciated that our method “allows editing of stereo audio direction” and concluded that the paper is “well-substantiated and advances reasoning-guided audio manipulation.”
>
>
> Despite the majority of positive ratings, including a score of 8, we have carefully considered all reviewer concerns and made revisions to strengthen the clarity, thoroughness, and presentation of the work. We highlight the major themes raised across reviews and describe how we addressed them.
>
>
> **1. Clarification of dataset creation and Strengthened Evidence for Real-World Generalization**
>
> One reviewer (QgiL) misunderstood that our dataset was built by “stitching” audio clips sequentially. We clarified that all mixtures are generated using physical linear superposition of real-world recordings. We introduce random time offsets to create natural overlap. Spatialization and reverberation are rendered using PyRoomAcoustics. To demonstrate the real-world generalization, we added some extra demo from raw Youtube recordings. Besides, we also clarify that all source audio clips already contain real-world noise, overlaps, and reverberation.
>
> **2. Expanded Editing Vocabulary and Spatial Controls and Declarative Editing dataset.**
>
> To address concerns from Reviewer eq8D and NKMZ, we added new atomic operations including time shift, add reverberation, timbre adjustments by scaling up the single-step audio editing dataset to 1M. Beyond this, spatial control is expanded up to more azimuth categories. We evaluated these new operations with baselines and included examples in the revised table. Beyond this, we also scaled up the declarative dataset from 50k to 250k and have some preliminary results showing that SmartDJ is still better than Audit. We also updated some demos for the new dataset and operations.
>
> **3. Quantitative Evaluation of ALM decomposition**
>
> Reviewers requested metrics supporting the ALM’s reasoning accuracy. We now include BERTScore evaluation of ALM-generated edit steps vs. ground truth and achieve a  F1 of 91.5%. We also provide additional examples and failure cases illustrating ALM behavior on abstract and conflicting instructions.
>
> **4. Demonstrating Why Captioning + LLM Cannot Replace the ALM and the Need for ALM**
>
> The reviewer (QgiL) was asking whether the ALM could be replaced by a captioning model plus a strong LLM. We added experiments and showed that captioning models miss fine-grained events, causing cascading reasoning errors in multi-step editing. We test two pipelines using (a) a state-of-the-art audio captioner and (b) AF2 as a captioner both underperform our ALM. Compared with these two alternatives, the fully fine-tuned AF2 consistently achieves the best results. This shows the importance of the ALM model and can not be replaced by an audio caption model + LLM.
>
> **5. Additional Baselines and Comparative Studies**
>
> We expanded our experimental section to also include zero-shot audio editing method on the “change spatial direction” operation. SmartDJ shows better performance compared to them. We also ablate the choice of ALM backbone where we fine-tune the LTU to compare with AF2 and show that LTU is also capable of declarative audio editing but has worse performance compared to AF2.
>
>
> **6. More Implementation Details & Manuscript Clarifications**
>
> Based on reviewer feedback, we revised: the data generation pipeline and room acoustics modeling and spatial audio rendering details. Human verification procedures for the evaluation set. These improvements enhance clarity, reproducibility, and readability.
>
> Thank you again for your time and for overseeing the review process. We are happy to provide any additional clarification.

---

> ### Author Response · Authors · 2025-11-21
> **Author Response (2/2)**
>
> ---
> > **Expanding the Current Atomic Operations: “Are there any additional atomic operations to consider beyond the current ones?”**
>
> Thank you for your suggestion on expanding the operations. We have expanded the editing vocabulary of SmartDJ to support more expressive manipulations. Specifically, our stereo-aware editor adds the following operations in addition to add, remove, extract and direction/volume adjustment in the original paper:
> - Time shift: adjusting the sound timing by x seconds
> - Add reverberation to a single source source:
>     - Low reverberation level: convolve with RIR with RT60 of 0.4s
>     - Mid reverberation level: convolve with RIR with RT60 of 0.8s
>     - High reverberation level: convolve with RIR with RT60 of 1.2s
> - Timbre adjust:
>     - Bright: boost high frequency >3kHz
>     - Dark: reduce high frequency >3kHz
>     - Warm: boost components between 300Hz~4kHz and
>     - Cold: boost components >6kHz and reduce components <300Hz
>     - Muffled: reduce the component >1.5kHz
>
>
> These extensions enable SmartDJ to handle more diverse and broader editing scenarios beyond our previous operations. We also evaluate these new operations in the following table. SmartDJ consistently improves over the Audit baseline. Additionally, we provide new examples of these operations in the updated project page in the supplementary materials.
>
> | Operation  | Method  |   FD  |  FAD |  KL  |  LSD |  GCC  |  CRW  | FSAD |
> |------------|---------|:-----:|:----:|:----:|:----:|:-----:|:-----:|:----:|
> | Shift time |  Audit  | 31.91 | 5.11 | 1.35 | 1.31 | 127.9 | 125.9 | 0.87 |
> |  | SmartDJ | 10.01 | 1.25 | 0.40 | 1.12 |  1.82 | 25.96 | 0.10 |
> |   Timbre   |  Audit  | 31.71 | 3.91 | 1.40 | 1.09 | 135.2 | 135.9 | 0.84 |
> |  | SmartDJ |  8.68 | 0.80 | 0.35 | 1.04 |  3.87 |  32.2 | 0.11 |
> |   Reverb   |  Audit  | 31.42 | 3.68 | 1.45 | 1.06 | 132.1 | 127.5 | 0.81 |
> |   | SmartDJ | 10.20 | 0.98 | 0.47 | 1.01 |  6.67 | 23.16 | 0.12 |
>
>
> ---
> > **Explanation on the Subjective Preference between SmartDJ and Audit: “When using Audit as a baseline .. SmartDJ and Audit in a single-step test?”**
>
> Both SmartDJ and Audit are trained on exactly the same dataset. The strong subjective preference for SmartDJ is consistent with the objective metric results and is explained by two major differences in model design:
>
> -  Higher-fidelity audio representation. Audit’s VAE operates on mel-spectrograms and discard phase information. The phase information is a crucial element for preserving spatial cues in stereo audio. In contrast, SmartDJ employs a two-channel waveform-based audio autoencoder. This VAE model encodes the full stereo waveform directly, which preserves spatial information and achieves higher-fidelity reconstruction.
>
> - Stronger generative backbone. Audit uses a U-Net diffusion model, while SmartDJ employs a Diffusion Transformer, which models long-range temporal structure better and provides richer cross-attention-based text conditioning.
>
> As a result, even with identical training data, SmartDJ’s stronger representation and editing backbone lead to both objectively better edits and higher subjective preference.

---

> > ### Comment · Reviewer_eq8D · 2025-11-25
> >
> > Thank you to the authors for responding to the questions and concerns, and for offering more details and explanations. I would like to maintain a score of 6.

---

> > > ### Author Response · Authors · 2025-11-26
> > >
> > > Thank you very much for your time, thoughtful feedback, and for maintaining your positive score. We truly appreciate your careful evaluation and comments throughout the review process

---

### Official Review · Reviewer_bDhL · 2025-11-01

**Soundness:** 3
**Presentation:** 3
**Contribution:** 3
**Rating:** 8
**Confidence:** 4

**Summary:**

This paper introduces SmartDJ, a novel framework for declarative stereo audio editing. It leverages an Audio Language Model (ALM) to decompose high-level user instructions (e.g., "make this sound like a sunny forest") into a sequence of atomic editing operations (e.g., add/remove sounds, adjust volume/direction). These operations are then executed sequentially by a Latent Diffusion Model (LDM) to edit the audio while preserving spatial realism.

The authors also propose a scalable data synthesis pipeline using LLMs and signal processing to generate paired training data, addressing the lack of such datasets. Key contributions include: (1) the first declarative stereo audio editor combining ALM reasoning with LDM generation; (2) a controllable data pipeline for editable audio scenes; and (3) empirical results showing superior perceptual quality, semantic alignment, and spatial fidelity over baselines like Audit and WavCraft, validated through metrics (e.g., FD, CLAP, FSAD) and human studies.

**Strengths:**

The paper demonstrates strong originality by adapting multimodal large language models (MLLMs) and diffusion-based editing techniques from the vision domain to audio, specifically addressing the underexplored area of declarative (high-level) stereo audio editing.

The integration of an ALM as a "planner" for decomposing instructions into atomic steps, combined with an LDM "editor," is a clever modular design that allows for interpretability and potential human-in-the-loop refinement. The data synthesis pipeline is a standout contribution: using GPT-4o as a "designer" to generate instructions and steps, paired with rule-based audio composition, enables scalable training data creation, which is a practical solution to the data scarcity problem in audio editing.

Experiments are comprehensive, including quantitative metrics (e.g., lower FD and higher CLAP scores in Table 1), ablations (e.g., multi-round editing quality in Table 3), and human evaluations showing preference for SmartDJ in fidelity and alignment. Overall, the work is well-substantiated and advances reasoning-guided audio manipulation.

**Weaknesses:**

While the synthetic data pipeline is innovative, the heavy reliance on composed audio from clean datasets (e.g., AudioCaps, FSD50k) may limit generalization to real-world recordings, which often include overlapping events, noise, or reverberation not fully captured in the synthesis. For instance, the atomic operations are restricted to a predefined set (add/remove/extract/volume/direction), which might not handle more complex edits like timbre changes or temporal alignments, potentially reducing flexibility for diverse instructions.

Computationally, the sequential multi-step editing (e.g., 13.1s inference time in Table 1) is slower than end-to-end baselines, which could hinder real-time applications.

The baselines are appropriate but could be expanded to include more recent audio editing methods (e.g., adaptations of IP2P or P2P from images to audio, beyond just Audit and WavCraft). Human studies are mentioned but details (e.g., number of participants, inter-rater agreement) are sparse in the main text, making it hard to assess robustness without appendices.

**Questions:**

1. How well does SmartDJ generalize to real-world audio recordings (e.g., field recordings with ambient noise or overlaps) versus the synthetic mixtures used in training? Have you tested this, and if not, what challenges do you anticipate?

2. The ALM decomposes instructions effectively in your examples, but how does it handle ambiguous, abstract, or conflicting high-level prompts (e.g., "make it eerie yet cheerful")? Could you provide failure cases or metrics for decomposition accuracy?

3. Are there plans to expand the set of atomic operations (e.g., to include reverb, pitch shifting, or event timing adjustments) to support broader editing scenarios?

4. In the human evaluation, how many participants were involved, what was the setup (e.g., A/B preference, rating scales), and what was the statistical significance of the preferences? This could strengthen the subjective claims.

5. Given the modular design, how sensitive is the overall performance to the choice of ALM backbone (e.g., LTU vs. Audio Flamingo)? Did you ablate different ALMs?

---

> ### Author Response · Authors · 2025-11-21
> **Author Response (1/2)**
>
> We would like to thank you for your appreciation of the work. We are glad that you found our work “demonstrates strong originality” and “experiments are comprehensive”.  We address your concerns as follows:
>
> ---
> > **Dataset and Real-World Generalization: “How well does SmartDJ generalize to real-world … What challenges do you anticipate?”**
>
> We would first like to clarify that the raw audio clips used in the paper are not ideal audios. They come from real-world recordings that contain noise and reverberations. They come from real-world recordings. These sources naturally contain background noise, overlaps, and varying levels of reverberation. Moreover,  when multiple raw clips are mixed, the resulting audio could contain more overlapping audio events.
>
> To further assess real-world generalization, we additionally tested SmartDJ on raw YouTube recordings (included in the supplementary material). In these cases, the ALM produced coherent multi-step plans and the LDM executed the edits reliably. This suggests that the system transfers reasonably well beyond the synthetic mixtures.
>
> However, more challenging real-world conditions, including heavy noise, handheld motion, or strong domain shift, remain challenging for any generative editor. Improving robustness in these scenarios would likely require training on noisier or more diverse data, and we view this as an important direction for future work.
>
> ---
> > **Working with more Atomic Operations: “Are there plans to expand … broader editing scenarios?”**
>
> Thank you for your suggestion on expanding the operations. We have expanded the editing vocabulary of SmartDJ to support more expressive manipulations. Specifically, our stereo-aware editor adds the following operations in addition to add, remove, extract and direction/volume adjustment in the original paper:
> - Time shift: adjusting the sound timing by x seconds
> - Add reverberation to a single source source:
>     - Low reverberation level: convolve with RIR with RT60 of 0.4s
>     - Mid reverberation level: convolve with RIR with RT60 of 0.8s
>     - High reverberation level: convolve with RIR with RT60 of 1.2s
> - Timbre adjust / Sound color adjust:
>     - Bright: boost high frequency >3kHz
>     - Dark: reduce high frequency >3kHz
>     - Warm: boost components between 300Hz~4kHz and
>     - Cold: boost components >6kHz and reduce components <300Hz
>     - Muffled: reduce the component >1.5kHz
>
> These additional operations allow SmartDJ to handle more diverse editing scenarios beyond our previous operations. We also evaluate these new operations with the results in the following table. SmartDJ consistently improves over the Audit baseline. Additionally, we provide new examples of these operations in the updated project page in supplementary materials.
>
> | Operation  | Method  |   FD  |  FAD |  KL  |  LSD |  GCC  |  CRW  | FSAD |
> |------------|---------|:-----:|:----:|:----:|:----:|:-----:|:-----:|:----:|
> | Shift time |  Audit  | 31.91 | 5.11 | 1.35 | 1.31 | 127.9 | 125.9 | 0.87 |
> | | SmartDJ | 10.01 | 1.25 | 0.40 | 1.12 |  1.82 | 25.96 | 0.10 |
> |   Timbre   |  Audit  | 31.71 | 3.91 | 1.40 | 1.09 | 135.2 | 135.9 | 0.84 |
> | | SmartDJ |  8.68 | 0.80 | 0.35 | 1.04 |  3.87 |  32.2 | 0.11 |
> |   Reverb   |  Audit  | 31.42 | 3.68 | 1.45 | 1.06 | 132.1 | 127.5 | 0.81 |
> | | SmartDJ | 10.20 | 0.98 | 0.47 | 1.01 |  6.67 | 23.16 | 0.12 |
>
>
> ---
> > **Inference Speed: “Computationally, the sequential multi-step editing … real-time applications.”**
>
> We acknowledge that multi-step editing introduces additional inference time. Generating one set of atomic instructions via the ALM takes ~4.8s, and a full multi-round edit takes ~13s for a 10s clip. This is indeed slower than an end-to-end LDM. However, this modular design has many benefits: (1) the step-wise control enables human-in-the-loop editing and interaction; (2) it decouples the reasoning from editing, making the system far more sample-efficient and achieving better performance. In contrast, an end-to-end editor requires substantially more paired training data to cover the full space of complex instructions. Our modular design alleviates this data requirement.
>
> In addition, we can integrate recent fast diffusion methods in future work, such as SoundCTM [1] and Presto [2], which can process a 5 second audio clip within 0.1 second. Combining these frameworks in the future can make the full pipeline achieve faster inference speed.
>
> [1] SoundCTM: Unifying Score-based and Consistency Models for Full-band Text-to-Sound Generation. ICLR 2025.
>
> [2] Presto! Distilling Steps and Layers for Accelerating Music Generation. ICLR 2025.

---

> ### Author Response · Authors · 2025-11-21
> **Author Response (2/2)**
>
> ---
> > **Adapt Baselines from Image Domain: “The baselines are … (e.g., adaptations of IP2P or P2P from images to audio)”**
>
> We would like to clarify that our evaluation baselines include the audio adaptations of these specific architectures.
>
> - Standard Inversion (DDIM/DDPM): We included baseline inversion methods ZETA, SDEdit, and standard DDIM Inversion that primarily focus on editing the input signal with global noise-mixing.
>
> - Prompt-to-Prompt (P2P): Our AudioEditor baseline represents P2P. The method implements P2P's core contribution, Cross-Attention Control, and a further Null-Text Inversion technique.
>
> - InstructPix2Pix (IP2P): IP2P is similar to our end-to-end training strategy. Our results show that this method doesn’t perform well since the diffusion model is difficult to reason about declarative edits.
>
> ---
> > **Details of User Study: “In the human evaluation, … This could strengthen the subjective claims.”**
>
> We conducted a blind, pairwise A/B preference survey with 19 participants. Each participant evaluated 20 randomly sampled pairs (10 declarative editing, 10 single-step editing). In each pairwise comparison, the participants were asked to select the audio with higher quality and better alignment to instructions between our method and one of the baselines.
>
>
> Regarding the statistical significance of the preferences, we evaluated the win rate of SmartDJ against three baselines. Significance was calculated using a two-sided Z-test against a null hypothesis of random chance (p=0.5), and we report 95% Confidence Intervals (CI) to demonstrate the stability of the preferences.
>
> We will revise the manuscript by adding more details and results.
>
> | Task & Metric | Baseline | Preference (SmartDJ) | 95% Confidence Interval | p-value |
> |:--:|--|:--:|:--:|:--:|
> | Declarative Edit: Quality  | ZETA  |  80.00% | [68.9% – 91.1%]  | < 0.001 |
> || AudioEditor |  95.50% | [90.6% – 100.0%]  | < 0.001 |
> || Audit |  90.40% | [83.7% – 97.2%]  | < 0.001 |
> |Declarative Edit: Alignment | ZETA |  87.00% | [80.4% – 93.6%]  | < 0.001 |
> | | AudioEditor |  91.00% | [86.2% – 95.9%]  | < 0.001 |
> || Audit | 93.20% | [89.1% – 97.2%] | < 0.001 |
> | Single Step: Quality | ZETA | 77.40% | [66.1% – 88.6%]   | < 0.001 |
> | | AudioEditor |  86.30% | [76.8% – 95.7%] | < 0.001 |
> || Audit | 95.30% | [90.9% – 99.8%]  | < 0.001 |
> |Single Step: Alignment | ZETA  |  88.90% | [83.4% – 94.4%]  | < 0.001 |
> | | AudioEditor |   84.90% | [78.7% – 91.2%]  | < 0.001 |
> | | Audit |  96.80% | [94.2% – 99.3%]  | < 0.001 |
>
> ---
> > **Failure Cases of ALM: “The ALM decomposes instructions… provide failure cases or metrics for decomposition accuracy?”**
>
> To better illustrate how the ALM behaves under abstract or conflicting prompts such as “make it sound eerie yet cheerful,” we conducted a test using this instruction across different audio inputs:
>
> - Example 1. The audio contents include child cries and wind blows. And ALM outputs: add laughing sound, add person whispers; turn up child cries. In this case, the ALM successfully adds person whispers and enhances child cries to satisfy the “eerie’’ aspect. To make it “cheerful,” it adds a laughing sound. In this case, the ALM can successfully edit the original audio to satisfy both the emotional transfer but the resulting audio scene might sound conflicting.
>
> - Example 2. The audio contents include female speech and bell ringing. And ALM outputs: add owl call; turn up woman speech. Here, the ALM again handles the “eerie’’ component reasonably by adding an owl call. It tried to enhance cheerfulness by turning up female speech, but it fails to produce edits that convey explicit “cheerful,” indicating difficulty with conflicting or abstract prompts.
>
> Overall, these examples show that the ALM can reliably handle atmosphere-related cues (e.g., “eerie”, “cheerful”), but it is still a challenging task when emotional or stylistic attributes conflict or lack clear instruction. This reflects an inherent limitation of mapping abstract emotions to concrete operations. Extending the ALM to better model emotional and stylistic intent is a natural direction for future work.
>
> ---
> > **Ablation on ALM backbone: “Given the modular design, … ablate different ALMs?”**
>
> To assess the sensitivity of SmartDJ to the choice of ALM backbone, we replaced our AF2-based ALM with LTU while keeping the editor unchanged. As shown in the table, using LTU results in slightly worse performance. We attribute this gap primarily to the fact that our ALM is pretrained on a much larger and more diverse corpus, giving it stronger performance on decomposing instructions. Nevertheless, the LTU+SmartDJ editor variant still produces reasonable results. This indicates that our framework is generalizable to different model ALM architecture.
> |  Variant | FD | FAD| KL |  LSD | IS | CLAP|
> |:-----:|:--:|:---:|:--:|:--:|:--:|:--:|
> | LTU+SmartDJ editor | 16.5 | 1.58 | 3.00 | 1.79 | 8.10 | 0.204 |
> | SmartDJ | 14.7 | 1.53 | 2.85 | 1.42 | 8.36 | 0.238 |

---

### Author Response · Authors · 2025-12-03
**Rebuttal summary**

Dear Area Chair and Reviewers,

We sincerely thank the Area Chair for overseeing the review process and the reviewers for thoughtful assessments and overall positive evaluations. Below we first summarize the key comments from the reviewers, followed by the major revisions we have made to address the concerns.

Our paper’s strengths were consistently highlighted across three main dimensions: novelty of the task, originality of the methodology, and strong empirical capabilities and results.

**1. Novelty of the Task**

Reviewer QgiL noted that our work “addresses a novel and forward-looking task,” emphasizing that declarative and stereo-audio editing is “technically more challenging and rarely explored in prior works.” Reviewer NKMZ similarly remarked on our “novel task formulation” and “valuable dataset contribution.”


**2. Originality of our Method**

Reviewer QgiL highlighted the “creative” nature of our approach and its potential to “inspire future work combining reasoning and audio generation.” Reviewer bDhL further praised the “strong originality” and “clever modular design,” a sentiment echoed by reviewer eq8D, who commended our “innovative use of ALM and LLM.”


**3. Strong capabilities and results**

Reviewer bDhL described our experiments as “comprehensive,” while reviewer eq8D appreciated that our method “allows editing of stereo audio direction” and concluded that the paper is “well-substantiated and advances reasoning-guided audio manipulation.”


Despite the majority of positive ratings, including a score of 8, we have carefully considered all reviewer concerns and made revisions to strengthen the clarity, thoroughness, and presentation of the work. We highlight the major themes raised across reviews and describe how we addressed them.


**1. Clarification of dataset creation and Strengthened Evidence for Real-World Generalization**

One reviewer (QgiL) misunderstood that our dataset was built by “stitching” audio clips sequentially. We clarified that all mixtures are generated using physical linear superposition of real-world recordings. We introduce random time offsets to create natural overlap. Spatialization and reverberation are rendered using PyRoomAcoustics. To demonstrate the real-world generalization, we added some extra demo from raw Youtube recordings. Besides, we also clarify that all source audio clips already contain real-world noise, overlaps, and reverberation.

**2. Expanded Editing Vocabulary and Spatial Controls and Declarative Editing dataset.**

To address concerns from Reviewer eq8D and NKMZ, we added new atomic operations including time shift, add reverberation, timbre adjustments by scaling up the single-step audio editing dataset to 1M. Beyond this, spatial control is expanded up to more azimuth categories. We evaluated these new operations with baselines and included examples in the revised table. Beyond this, we also scaled up the declarative dataset from 50k to 250k and have some preliminary results showing that SmartDJ is still better than Audit. We also updated some demos for the new dataset and operations.

**3. Quantitative Evaluation of ALM decomposition**

Reviewers requested metrics supporting the ALM’s reasoning accuracy. We now include BERTScore evaluation of ALM-generated edit steps vs. ground truth and achieve a  F1 of 91.5%. We also provide additional examples and failure cases illustrating ALM behavior on abstract and conflicting instructions.

**4. Demonstrating Why Captioning + LLM Cannot Replace the ALM and the Need for ALM**

The reviewer (QgiL) was asking whether the ALM could be replaced by a captioning model plus a strong LLM. We added experiments and showed that captioning models miss fine-grained events, causing cascading reasoning errors in multi-step editing. We test two pipelines using (a) a state-of-the-art audio captioner and (b) AF2 as a captioner both underperform our ALM. Compared with these two alternatives, the fully fine-tuned AF2 consistently achieves the best results. This shows the importance of the ALM model and can not be replaced by an audio caption model + LLM.

**5. Additional Baselines and Comparative Studies**

We expanded our experimental section to also include zero-shot audio editing method on the “change spatial direction” operation. SmartDJ shows better performance compared to them. We also ablate the choice of ALM backbone where we fine-tune the LTU to compare with AF2 and show that LTU is also capable of declarative audio editing but has worse performance compared to AF2.


**6. More Implementation Details & Manuscript Clarifications**

Based on reviewer feedback, we revised: the data generation pipeline and room acoustics modeling and spatial audio rendering details. Human verification procedures for the evaluation set. These improvements enhance clarity, reproducibility, and readability.

Thank you again for your time and for overseeing the review process. We are happy to provide any additional clarification.

---

### Meta-Review · Area_Chair_VFQT · 2026-01-05

**Summary:**

Reviewers generally agreed that the paper addresses an interesting task: declarative stereo audio editing. SmartDJ is proposed in this work, which is a framework that decomposes abstract instructions into a sequence of atomic editing operations executed by a LDM. Strong results are achieved as compared with the baselines.

The initial main concerns raised related to dataset, novelty, and evaluation, the majority of which were satisfactorily addressed in the rebuttal. Based on the overall assessments, the Area Chair recommends Accept (poster).

**Reviewer Concerns:**

Most of the concerns were addressed by the rebuttal, while Reviewer QgiL remains unconvinced about novelty of this work and raises concerns about its real-world generalization .

**Reviewer Scores:**

I think the reviewer scores are unlikely to change after discussion.

---

### Decision · Program_Chairs · 2026-01-26

Accept (Poster)